# Bridging the gap between single-cell migration and collective dynamics

**Florian Thüroff[†], Andriy Goychuk[†], Matthias Reiter, Erwin Frey\***

Arnold Sommerfeld Center for Theoretical Physics and Center for NanoScience, Department of Physics, Ludwig-Maximilians-Universität München, Munich, Germany

**Abstract** Motivated by the wealth of experimental data recently available, we present a cellular-automaton-based modeling framework focussing on high-level cell functions and their concerted effect on cellular migration patterns. Specifically, we formulate a coarse-grained description of cell polarity through self-regulated actin organization and its response to mechanical cues. Furthermore, we address the impact of cell adhesion on collective migration in cell cohorts. The model faithfully reproduces typical cell shapes and movements down to the level of single cells, yet allows for the efficient simulation of confluent tissues. In confined circular geometries, we find that specific properties of individual cells (polarizability; contractility) influence the emerging collective motion of small cell cohorts. Finally, we study the properties of expanding cellular monolayers (front morphology; stress and velocity distributions) at the level of extended tissues.

**\*For correspondence:**
frey@lmu.de

[†]These authors contributed equally to this work

**Competing interests:** The authors declare that no competing interests exist.

## Introduction

Cell movements range from uncoordinated ruffling of cell boundaries to the migration of single cells (*Ridley et al., 2003*) to the collective motions of cohesive cell groups (*Friedl and Gilmour, 2009*). Single-cell migration enables cells to move towards and between tissue compartments – a process that plays an important role in the inflammation-induced migration of leukocytes (*Friedl and Weigelin, 2008*). One can distinguish between amoeboid and mesenchymal migration, which are characterized by widely different cell morphologies and adhesive interactions with their respective environments (*Friedl, 2004*; *Lämmermann and Sixt, 2009*). Cells may also form cohesive clusters and mobilize as a collective (*Trepat et al., 2009*; *Angelini et al., 2011*; *Doxzen et al., 2013*; *Deforet et al., 2014*; *Vedula et al., 2012*; *Marel et al., 2014*). This last mode of cell migration is known to drive tissue remodelling during embryonic morphogenesis (*Lecaudey and Gilmour, 2006*) and wound repair (*Poujade et al., 2007*).

Despite this broad diversity of migration modes, there appears to be a general consensus that all require (to varying degrees) the following factors: (i) Cell polarization, cytoskeletal (re)organization, and force generation driven by the interplay between actin polymerization and contraction of actomyosin networks. (ii) Cell-cell cohesion and coupling mediated by adherens-junction proteins which are coupled to the cytoskeleton. (iii) Guidance by chemical and physical signals. The basic functionalities implemented by these different factors confer on cells the ability to generate forces, adhere (differentially) to each other and to a substrate, and respond to mechanical and chemical signals. However, a fully mechanistic understanding of how these basic functionalities are integrated into single-cell migration and coordinated multicellular movement is still lacking.

Here, we present a computational model which enables us to study cell migration at various scales, and thus provides an integrative perspective on the basic cell functions that enable the emergence of collective cell migration. While a variety of very successful modeling approaches has been used to describe single-cell dynamics (*Mogilner, 2009*; *Marée et al., 2006*; *Marée et al., 2012*; *Shao et al., 2010*; *Ziebert et al., 2012*; *Ziebert and Aranson, 2013*; *Camley et al., 2013*; *Albert and Schwarz, 2014*; *Dietrich et al., 2018*; *Goychuk et al., 2018*) or the movements of

extended tissues (*Szabó et al., 2006*; *Szabó et al., 2010*; *Kabla, 2012*; *Sepúlveda et al., 2013*; *Basan et al., 2013*; *Banerjee et al., 2015*; *Alt et al., 2017*; *Tarle et al., 2017*), these models are hard to reconcile with each other. Models that focus on single cells are typically difficult to extend to larger cell numbers, largely due to their computational complexity. On the other hand, approaches which are designed to capture the dynamics at the scale of entire tissues generally adopt a rather coarse-grained point of view, and are therefore difficult to transfer to single cells or small cell cohorts. At present there are two partly competing and partly complementary approaches to bridge the gap between single-cell migration and collective dynamics, namely phase-field models (*Shao et al., 2010*; *Ziebert et al., 2012*; *Shao et al., 2012*; *Camley et al., 2014*; *Camley and Rappel, 2014*; *Löber et al., 2015*), and cellular Potts models (CPMs) (*Szabó et al., 2010*; *Kabla, 2012*; *Szabó and Merks, 2013*; *van Oers et al., 2014*; *Segerer et al., 2015*; *Niculescu et al., 2015*; *Albert and Schwarz, 2016*; *Rens and Merks, 2017*) first introduced by *Graner and Glazier (1992)*.

## Box 1. A simple description of complex cells?

Mammalian cells are made up of around $10^9$ interacting proteins (*Milo and Phillips, 2015*) in an aqueous compartment enclosed by a lipid bilayer membrane. A substantial fraction of these proteins is devoted to the structural support of the cell. The cytoskeletal systems that perform this function also mediate elastic deformations of the cell through stresses induced by motor proteins. Cell migration is enabled by transient, transmembrane attachment of the cytoskeleton to external structures (extracellular matrix or a substrate) via integrins, and regulated by various signaling pathways. To gain insights into such a complex system, we simplify these networks, each comprised of many interacting components, into coarse building blocks, which might seem arbitrary at first, but serve to qualitatively capture generic features of the underlying machinery. These generic and qualitative building blocks allow us to finally arrive at a quantitative description of cell dynamics.

Building on and generalizing the CPM (*Graner and Glazier, 1992*), we present a cellular automaton model that is designed to capture essential cellular features even in the context of the migration of single cells and of small sets of cells. At the same time, it is computationally efficient for simulations with very large cell numbers (currently up to $\mathcal{O}(10^4)$ cells), thus permitting investigations of collective dynamics at the scale of tissues. Our model reproduces the most pertinent features of cell migration even in the limiting case of solitary cells, and is compatible with a wealth of experimental evidence derived from both small cell groups and larger collectives made up of several thousand cells. Specifically, by studying the characteristics of single-cell trajectories and of small cell groups confined to circular territories, we demonstrate that persistency of movements is significantly affected by cell stiffness and cell polarizability. Moreover, we investigate the dynamics of tissues in the context of a typical wound-healing assay (*Poujade et al., 2007*; *Trepat et al., 2009*; *Serra-Picamal et al., 2012*), and show that the model exhibits the recurring mechanical waves observed experimentally (*Serra-Picamal et al., 2012*), a feature which we attribute to the coupling between cell-sheet expansion and cell-density-induced growth inhibition.

## Computational model
### Model geometry
We consider cells that adhere to a two-dimensional surface, spanned by the coordinates $(x, y)$, through some contact area (*Figure 1A*). Membrane protrusions and retractions, which determine cell motion and shape (*Pollard and Borisy, 2003*; *Lauffenburger and Horwitz, 1996*), correspond to size and shape changes of the surface contact area. We assume that processes that take place at the cell boundary drive cell motion, and therefore disregard the cell body, which extends into the $z$-direction. In our computational model, we tesselate the available surface into a honeycomb lattice, where each hexagon corresponds to a discrete adhesion between the cell and the substrate. Then, protrusion and retraction events correspond to the gain and loss of hexagons at the boundary of the substrate contact area, respectively. The occurrence of these events is determined by a Monte Carlo scheme gradually minimizing an effective energy, $\mathcal{H}$, which is associated with the cell configuration.

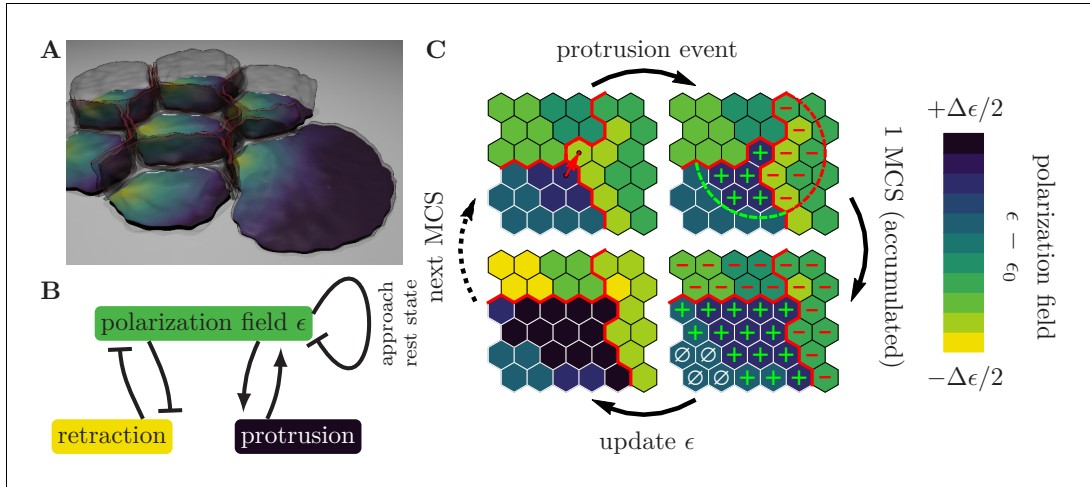

**Figure 1.** Illustration of the computational model with the pertinent simulation steps. (A) Illustration of a small cell cohort that adheres to a surface ($(x, y)$-plane). The polarization field, $\epsilon$, is defined on the contact surface with the adhesion plane. The magnitude of the polarization field, which is indicated by the colorbar in Figure (C), encodes the local strength of cell-substrate adhesions and emulates the local mass of force-generating (pushing) cytoskeletal structures. Cell-cell adhesions are indicated in red. (B) Cytoskeletal structures respond to external mechanical stimuli through reaction networks involving different feedback loops. We greatly simplify these complex processes into two prototypic feedback loops, which break detailed balance and drive cell migration, as follows. The polarization field induces membrane protrusions and inhibits retractions. In turn, protrusions increase the polarization field (positive feedback) and therefore the likelihood of further protrusive activity, while retractions decrease the polarization field (negative feedback). In the absence of mechanochemical signals, the polarization field approaches its rest state. (C) Zoom-in to a common boundary shared between the substrate contact areas of three cells (bounded by the red lines), each represented by a contiguous set of occupied grid sites (hexagons). *Top left:* The upper right corner of the lower left cell (source cell) initiates a protrusion event against a neighboring element in the cell to its right (target cell), as indicated by the arrow, in an attempt to displace it. The success of each such attempted elementary event depends on the balance between contractile forces, cytoskeletal forces, and cell adhesion. *Top right:* If the protrusion event is successful, then the levels of regulatory factors are increased (decreased) in integer steps, at all lattice sites inside the source (target) cell that lie within a radius $R$ of the accepted protrusion event (as indicated by the plus and minus signs). *Bottom right:* During the course of one MCS, different levels of regulatory factors accumulate locally within each cell, with positive levels of regulatory factors (green plus signs) promoting a build-up of cytoskeletal structures, negative levels of regulatory factors (red minus signs) causing degradation of cytoskeletal structures, and neutral levels of regulatory factors (white zero signs) causing relaxation towards a resting state, as indicated in the *lower left image*. The color code indicates local levels of cytoskeletal structures, $\epsilon$.

The cell is perpetually driven out of equilibrium by active reorganization of its actomyosin network and focal adhesions.

## Coarse-grained cellular mechanics

As discussed above, the configuration of a cell at any given time $t$ is associated with a substrate contact area $A(t)$ and perimeter $P(t)$. We assume that the membrane and cortex deformations of each cell are constrained by the elastic energy

$$\mathcal{H}_{\text{cont}}(t) = \kappa_A A^2(t) + \kappa_P P^2(t), \tag{1}$$

where $\kappa_A$ and $\kappa_P$ are cell-type-specific stiffness parameters, similar to the original implementation of the CPM (*Graner and Glazier, 1992*). If the cell does not form adhesions to the substrate, then membrane and cortex contractility will round up the cell body, thereby collapsing the substrate contact area into a contact point.

## Gripping the surface through the cell cytoskeleton

Detachment of the cell from the substrate is counteracted by focal adhesions, where the cell cytoskeleton is connected to the underlying substrate by integrins. Cellular protrusions are driven by outward pushing forces generated by the assembly and disassembly of cytoskeletal structures (*Pollard and Borisy, 2003*; *Mogilner, 2009*). As a first approximation, we subsume all of these complex dynamic processes, like the formation/degradation of focal adhesions and the assembly/disassembly of cytoskeletal structures, into a single time-dependent and spatially resolved internal field for each cell, $\epsilon(\mathbf{x}, t)$. This *polarization field* emulates the mass of force-generating cytoskeletal structures in the associated hexagon, at position $\mathbf{x}$, which results in an effective, locally regulated, adhesion energy between cell and substrate. Consequently, the total energy associated with this polarization field is given by

$$\mathcal{H}_{\text{cyto}}(t) = -\sum_{\mathbf{x}} \epsilon(\mathbf{x}, t). \tag{2}$$

The polarization field must vanish at positions that are not occupied by a cell. Therefore, a retraction is associated with an energy penalty due to the loss of a substrate adhesion. Consequently, a protrusion, where one source hexagon 'conquers' a nearby target hexagon, is associated with an energy gain due to an increase of the substrate contact area. Here, we assume that the newly incorporated hexagon has the same polarization field as its conqueror.

There are several biological factors that constrain the local density of actin filaments, myosin and focal adhesions, whose limited availability corresponds to an upper bound on the polarization field. Furthermore, we assume that there is some minimal attachment energy associated with adhesions that prevents the cells from detaching from the substrate, which implies a lower bound on the polarization field. This motivates to introduce cell-type-specific bounds for the polarization field: $\epsilon(\mathbf{x}, t) \in [\epsilon_0 - \Delta\epsilon/2, \ \epsilon_0 + \Delta\epsilon/2]$, where $\epsilon_0$ is the average polarization field and $\Delta\epsilon$ is the maximum cell polarity.

## Active self-regulation of the cytoskeleton

Assembly and disassembly of cytoskeletal structures are controlled by a myriad of accessory proteins (*Lauffenburger and Horwitz, 1996*; *Ridley et al., 2003*). These regulatory proteins form a reaction network involving different feedback mechanisms, which allow cytoskeletal structures to respond to external mechanical stimuli (*Marée et al., 2006*; *Marée et al., 2012*). Furthermore, cytoskeletal structures like integrins play a role in the spatiotemporal control of these regulatory proteins (*Schwartz and Shattil, 2000*). Here, we refrain from formulating a detailed reaction-diffusion model that accounts for the interactions between all of these contributing players. Instead, we assume that the internal chemistry of the cell will generically produce protein patterns, with a typical length scale $R$, which locally up- or down-regulate cellular cytoskeleton and focal adhesion (dis)assembly. Then, we greatly simplify these complex processes (*Lauffenburger and Horwitz, 1996*; *Schwartz and Shattil, 2000*; *Ridley et al., 2003*) into two prototypic feedback loops (*Figure 1B,C*):

A. The polarization field locally promotes outward motion of the membrane, because it contains a contribution from the local amount of actin filaments. Membrane protrusions facilitate the formation of substrate adhesions and further polymerization of actin filaments, leading to a positive feedback on the polarization field within a range $R$.

B. The polarization field also locally inhibits inward motion of the membrane, by emulating the local adhesion strength of the cell to the substrate. If a membrane retraction is successful, then the loss of substrate adhesions locally further increases cell contractility, leading to a negative feedback on the polarization field within a range $R$.

In the absence of regulatory signals, we assume that the polarization field decays to a fixed value, $\epsilon \to \epsilon_0$, which corresponds to a *resting state* of the cell cytoskeleton and focal adhesions. For the sake of keeping our model as simple as possible, we assume that all protein patterns have the same range $R$, and that the regulation of the cell cytoskeleton and focal adhesions follows a single timescale that corresponds to an *update rate* $\mu$. Because at heart, our model is only based on generic feedback loops with a certain signaling range $R$, we would argue that *any* model with similar feedback should, in general, lead to similar cell behavior. Indeed, mutually repressing feedback loops (*Marée et al., 2006*) and mutually activating feedback loops (*Shao et al., 2010*; *Ziebert et al.,*

*2012*; *Albert and Schwarz, 2016*) are crucial recurring motifs among multiple cell migration studies. Notably, these theoretical approaches all recover comparable cell behavior even when the model setup seems quite different at first glance:

1. Cell migration couples mechanochemically to a scalar field (*Shao et al., 2010*), if stresses in the cell are isotropic; this is analogous to the present study.
2. Cell migration couples mechanochemically to a vector field (*Marée et al., 2006*; *Ziebert et al., 2012*), if stresses in the cell are anisotropic.
3. Cell migration couples to a single polarity vector (*Albert and Schwarz, 2016*), if propulsive forces are distributed homogeneously throughout the cell. However, this simplification of the former two cases cannot account for the formation of multiple competing lamellopodia/pseudopods.

These different modeling approaches (of varying complexity) surprisingly yield a universal phenomenology. The puzzling similarity between these models suggests generic common features that determine cell shape and motility: mechanical constraints like cell elasticity and mechanochemical feedback mechanisms that break detailed balance, maintain cell polarity and drive cell motion.

## Intercellular adhesion and friction

In addition to internal remodeling of the cytoskeleton, adhesion of cells to neighboring cells and to the substrate plays a key role in explaining migratory phenotypes (*Mogilner, 2009*; *Friedl and Gilmour, 2009*). From a mechanical point of view, the implications of cell adhesion are two-fold:

1. Cell adhesion supports growth of cell-cell and cell-matrix contacts and may thus be described in terms of effective surface energies. In our computational model, cell-matrix contacts are readily accounted for by the polarization field, $\epsilon$. In addition, we associate the formation of cell-cell adhesions with an energy benefit $B$, which we call *cell-cell adhesion parameter*.
2. Once formed, adhesive bonds anchor the cell to the substrate and to neighboring cells. During cell migration, these anchoring points must continuously be broken up and reassembled (*Webb et al., 2002*; *Gumbiner, 2005*) and, hence, provide a constant source of energy dissipation. Therefore, we assume that the cost for rupturing an existing cell-cell adhesion, $B + \Delta B > B$, exceeds the gain from forming a new cell-cell adhesion. Then, the dissipative nature of cell-cell adhesions is accounted for by the *cell-cell friction parameter* $\Delta B$. Similarly, cell-matrix contacts can also provide a source of dissipation, which is further discussed in Appendix 2.

## Environmental cues

The polarization field, $\epsilon$, readily includes contributions from cell-substrate adhesions, which are locally up- or down-regulated by the cell. These cell-substrate adhesions require the abundance of surface ligands, which serve as substrate tethers that the cell can attach to, and which are not necessarily distributed homogeneously. By substrate micropatterning, one can arrange areas where the cell is likely to adhere to the surface, and *no-go-areas*, where the cell adheres less (or cannot adhere at all). To replicate such environmental cues, we introduce a second scalar field $\varphi(\mathbf{x})$, whose value is taken to reflect the relative availability of substrate sites at which focal adhesions between cell and substrate can be formed. Here, we have chosen to model micropatterns as impenetrable walls; we locally add a large energy penalty, $\varphi \ll 0$, to the polarization field ($\epsilon \rightarrow \epsilon + \varphi$), that a cell has to pay for trespassing onto a *no-go-area*. However, it is equally valid to treat $\varphi$ as a multiplicative constant modulating the polarization field ($\epsilon \rightarrow \varphi \epsilon$), where $\varphi = 0$ models a local inability of the cell to attach to the substrate. Analogously to cell-cell contacts, we account for the dissipative nature of cell-substrate adhesions by associating the breaking of such contacts with an additional energy cost $D$.

## Tissue growth by cell division

In the description so far, the cells are arrested in the cell cycle (mitostatic). To investigate the effect of cell proliferation on tissue dynamics, we introduce a simplified three-state model of cell division. Cells start off in a quiescent state, in which their properties remain constant over time. The cell sizes fluctuate around an average value determined by the cell properties and the local tissue pressure. Cell growth typically arrests at large cell densities, in a phenomenon coined *contact inhibition of proliferation* (*Stoker and Rubin, 1967*; *Puliafito et al., 2012*; *Pavel et al., 2018*). Since large cell

densities correspond to a small spread area for each individual cell, this implies that cell growth is arrested below a critical threshold size ($A_\mathrm{T}$). Upon exceeding this threshold size due to size fluctuations, cells leave the quiescent state and enter a growth state. The duration of the quiescent state is thus a random variable, whose average value depends on the tissue pressure, and lower pressure (due to a lower cell density) leads to a shorter quiescent state. During the subsequent deterministic growth state of duration $T_g$, cells double all of their cellular material and thus double in size. We model this growth as a gradual decrease in the effective cell contractility ($\kappa_A$ and $\kappa_P$). As there is no a priori reason to assume that a cell's migratory behavior should depend on its size, we constrain the parameters accordingly; this is described in detail in Appendix 2. After having grown for a duration $T_g$, cells switch to a deterministic division state of duration $T_d$. During division, cells strongly contract, which leads to mitotic rounding and a drastic decrease of their contact area with the substrate (*Jones et al., 2018*; *Lock et al., 2018*). In principle, a decrease of cell contact area could also lead to perturbations of the stress field in the monolayer. Here, however, we neglect the decrease of the cell spreading area, as the division phase is short compared to the growth phase. We expect that a drastic increase of cell contractility also leads to a loss of polarity in the cell's migratory machinery. Therefore, each cell reduces its polarizability to zero ($\Delta\epsilon \to 0$) in order to utilize its cytoskeleton for the separation of the cellular material, leading to mitotic rounding. At the end of the division state, each dividing cell splits into two identical daughter cells, whose properties and parameters are identical to the mother cell's initial values in the quiescent state. Finally, the daughter cells re-initialize migration from an unpolarized state. For a detailed and more technical description we refer the interested reader to Appendix 1.

## Results

### Persistent migration of single cells

The macroscopic properties of cell clusters and tissues emerge from an interplay between many individual cells. Then, what determines the mechanical and migratory features of these individual cells? In our computational model, we have studied this question by screening its multidimensional parameter space. For such a brute force approach to be numerically feasible, one must first distinguish relevant parameters (these determine the resulting dynamics) from irrelevant parameters. Specifically, in our extended cellular Potts model, there are *reference parameters* whose sole purpose is to control the spatial and temporal discretization of the numerical model:

1. The cytoskeletal update rate endows the cellular Potts model with a reference timescale and determines the temporal discretization. In this study, we have set $\mu = 0.1$.
2. The average polarization field $\epsilon_0$ encodes the energy gain for creating new cell-substrate adhesions, while the area stiffness $\kappa_A$ represents the energy cost for increasing the substrate contact area. Then, the number of hexagons occupied by the cell is proportional to the ratio $\epsilon_0/\kappa_A$. If we use a desired cell area as reference value, then the ratio $\epsilon_0/\kappa_A$ controls the spatial discretization of the cell. To study the migration of single cells and small cell cohorts, we have set the average polarization field to $\epsilon_0 = 225$ and the area stiffness to $\kappa_A = 0.18$.
3. In cellular Potts models, which are Monte-Carlo simulations, the reference energy of fluctuations is determined by an effective temperature. In this study, we have set $k_\mathrm{B}T \equiv 1$.

Furthermore, we used a large computational grid with $9 \cdot 10^4$ sites and periodic boundary conditions to study the migration of single cells. This leaves three parameters that control cell motility in the absence of cell-substrate dissipation: cell polarizability $\Delta\epsilon$, cell contractility $\kappa_P$ and signalling radius $R$. However, it is not clear yet whether all of these are independent relevant parameters. In fact, in the following sections it will become clear that cell polarizability and contractility are *degenerate parameters* (in the sense that the phenomenology only depends strongly on their ratio, which is the corresponding relevant parameter).

### Cell persistence increases with polarizability

First, we investigated the impact of varying levels of cell perimeter stiffness $\kappa_P$ and maximum cell polarity $\Delta\epsilon$ on the cell's migratory patterns (*Figure 2—video 1*), at a fixed signaling radius $R = 5$. To assess the statistics of the cell trajectories, we recorded the cell's orientation $\hat{\mathbf{v}}(t) \equiv \mathbf{v}(t)/\|\mathbf{v}(t)\|$ ($\mathbf{v}$: cell velocity) and (geometrical) center of mass position $\mathbf{R}(t)$ during a total simulation time of

$T_{\text{sim}} = 10^4$ Monte-Carlo steps (MCS). For each set of parameters, we performed 100 statistically independent simulations, from which we computed the mean squared displacement, $\text{MSD}(\tau) \equiv \langle [\mathbf{R}(t + \tau) - \mathbf{R}(t)]^2 \rangle$, and the normalized velocity auto-correlation function, $C(\tau) \equiv \langle \hat{\mathbf{v}}(t + \tau) \cdot \hat{\mathbf{v}}(t) \rangle$. Here, $\langle \ldots \rangle$ denotes an average with respect to simulation time $t$ as well as over all 100 independent simulations.

These computer simulations show that the statistics of the migratory patterns is well described by a *persistent random walk model* (*Stokes et al., 1991*; *Wu et al., 2014*) with its two hallmarks: a mean square displacement that exhibits a crossover from ballistic to diffusive motion (*Figure 2A*), and on sufficiently long time scales an exponential decay of the velocity autocorrelation function $C(\tau) \propto e^{-\tau/\tau_p}$ (inset of *Figure 2A*). We determined the persistence time of directed migration, $\tau_p$, by fitting the mean squared displacement with a persistent random walk model. In addition, we also measured cell speed, $v$, and cell aspect ratio, $l_+/l_-$, to further characterize cell motility and shape.

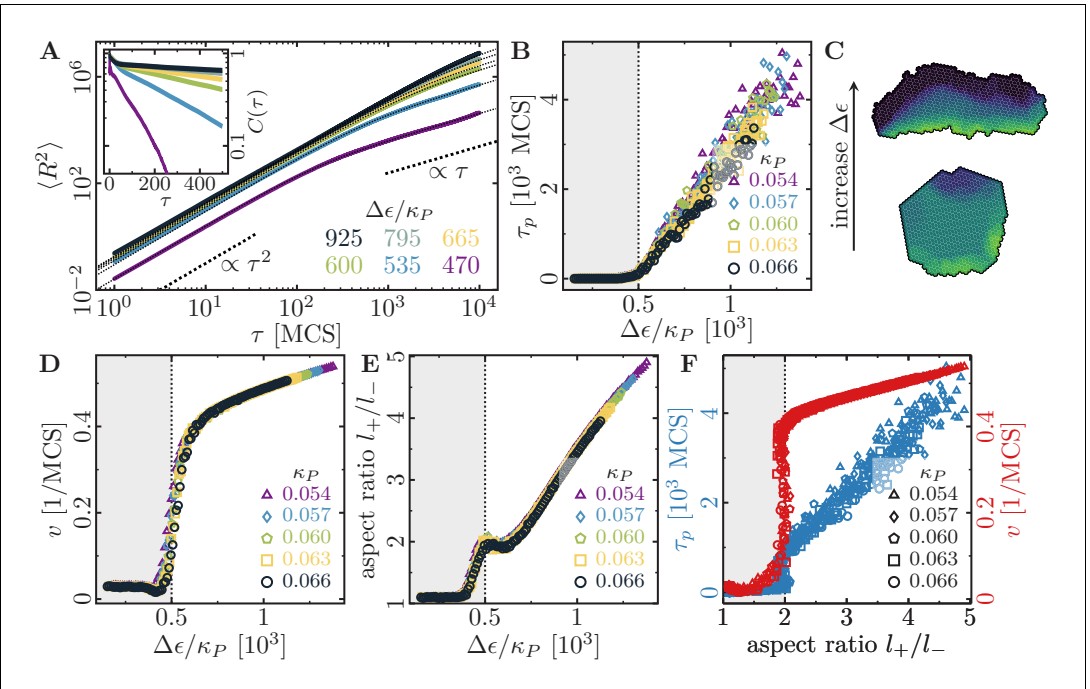

**Figure 2.** Cell shape and persistence of migration as a function of cell polarizability. (A) Mean-squared displacement (MSD) for single-cell movements at different maximum cell polarity $\Delta\epsilon$ (stiffness parameters $\kappa_P = 0.060$, $\kappa_A = 0.18$; average polarization field $\epsilon_0 = 225$; signaling radius $R = 5$; cell-substrate dissipation $D = 0$; cell-substrate adhesion penalty $\varphi = 0$; cytoskeletal update rate $\mu = 0.1$; 100 independent simulations for each set of parameters). Single cells perform a persistent random walk, i.e. they move ballistically (MSD $\propto \tau^2$) for $\tau \ll \tau_p$, and diffusively (MSD $\propto \tau$) for $\tau \gg \tau_p$. *Inset:* Normalized velocity auto-correlation function for the same parameters as in the main figure. (B) Persistence time of directed cell migration plotted as a function of maximum cell polarity $\Delta\epsilon$, and perimeter stiffness $\kappa_P$ (area stiffness $\kappa_A = 0.18$; average polarization field $\epsilon_0 = 225$; signaling radius $R = 5$; cell-substrate dissipation $D = 0$; cell-substrate adhesion penalty $\varphi = 0$; cytoskeletal update rate $\mu = 0.1$; 100 independent simulations for each set of parameters). The persistence time of the random walk increases with increasing cytoskeletal polarity and decreasing perimeter elasticity. (C) Cytoskeletal polarity also controls cell shapes, with crescent cell shapes (long persistence times) being observed at large cytoskeletal polarties, and round cell shapes (short persistence times) at small cytoskeletal polarities. Color code: cell polarization; cf. color bar in *Figure 1C*. (D) Single cell speed plotted as a function of maximum cell polarity $\Delta\epsilon$, and perimeter stiffness $\kappa_P$. (E) Single cell aspect ratio plotted as a function of maximum cell polarity $\Delta\epsilon$, and perimeter stiffness $\kappa_P$. (F) Speed and persistence time of single cells are correlated with the cell aspect ratio.

The online version of this article includes the following video and figure supplement(s) for figure 2:

**Figure supplement 1.** Role of substrate dissipation for cell shape and motility.

**Figure 2—video 1.** Single cell motility and shape for different maximum cell polarities ($\kappa_P = 0.060$, $R = 5$).

https://elifesciences.org/articles/46842#fig2video1

Surprisingly, for each of these variables we found a master curve that only depends on the ratio between cell polarizability and cell contractility, $\Delta\epsilon/\kappa_P$ (*Figure 2B,D,E*). This data collapse suggests $\Delta\epsilon/\kappa_P$ as a relevant parameter (while cell polarizability and contractility are degenerate parameters), which we will henceforth refer to as *specific polarizability*.

The cells' persistence times of directed migration, speeds and aspect ratios all show a characteristic dependence on the specific cell polarizability. There is a threshold value for the specific polarizability, $\Delta\epsilon/\kappa_P \approx 500$, below which cells remain immobile (*Figure 2B,D,E*; grey regions). Above this threshold, the persistence time of directed migration, speed and aspect ratio increase markedly with the specific polarizability (*Figure 2B,D,E*). In our model, the area and perimeter stiffnesses refer to global and homogeneous cell contractility, while the cell polarization field drives cell migration. As discussed in 'Gripping the surface through the cell cytoskeleton', the cell polarization field does not explicitly distinguish between a local extensibility (e.g. due to actin polymerization), a local contractility (due to myosin-induced contraction) of the cytoskeleton or spatially regulated cell-substrate adhesions. For example, if cell migration is driven by actin polymerization, then blebbistatin treatment will decrease the global cell contractility, which we predict to lead to more elongated cells that move faster and exhibit extended episodes of ballistic motion. Indeed, an increase of cell migration speed after blebbistatin treatment was observed for mouse hepatic stellate cells (*Liu et al., 2010*). Alternatively, cell migration could also be driven by myosin contractility, for example by pulling the cell forward or by locally detaching adhesions. Then, polarizability and contractility concomitantly depend on the ability of the cell to exert forces, which can be inhibited by blebbistatin treatment. If polarizability, $\Delta\epsilon$, and contractility, $\kappa_P$, are equally reduced by a blebbistatin-dependent prefactor, then the *specific polarizability*, $\Delta\epsilon/\kappa_P$, and the resulting cell phenomenology should remain unchanged. Indeed, blebbistatin treatment of keratocytes and keratocyte fragments was reported not to affect cell shape and speed to any significant degree (*Wilson et al., 2010*; *Ofer et al., 2011*). Therefore, blebbistatin treatment can either increase or decrease cell motility, depending on the cell type and possibly on the specific mechanism that drives cell migration.

Interestingly, because of this universal dependence of all the mentioned quantities on the specific polarizability, our simulations also show that there is a strong correlation between cell shape (aspect ratio) and cell motility (speed and persistence time of directed migration); see *Figure 2F*. While highly persistent trajectories are observed for cells with 'crescent' shapes, more erratic cell motion is typically found for cells with more rounded outlines (*Figure 2C*). In other words, our computational model predicts that cells which are able to polarize their cytoskeletal structures more strongly will adopt crescent shapes and show a higher degree of persistent cell motion. It would be interesting to further test these predictions by using phenotypic variations in cell shapes like those reported in experiments with keratocytes (*Keren et al., 2008*); there, the authors also found a correlation between cell shape and speed.

## Feedback range determines whether individual cells move persistently or rotate

Moreover, we investigated the influence of different signaling radii $R$ (typical range in which signalling molecules diffuse and mediate feedback mechanisms during a single Monte-Carlo step) on the persistence of single-cell trajectories. Since $R$ is the relevant parameter that controls the spatial organization of lamellipodium formation, its value should strongly affect the statistics of a cell's trajectory (*Figure 3A*). Indeed, at small values of $R$, we observe that the spatial coherence of cytoskeletal rearrangements is low, which frequently results in the disruption of ballistic motion due to the formation of independent lamellipodia in spatially separate sectors of the cell boundary (*Figure 3C*, lower snapshot). In contrast, at larger values of $R$, we find that spatial coherence is restored, and the formation of one extended lamellipodium across the cell's leading edge maintains a distinct front-rear axis of cell polarity (*Figure 3C*, upper snapshot). However, when the signaling radius is too large compared to the cell size, we find an inhibition of ballistic motion and rounding of the cells as signals originating from one cell edge begin to reach the opposing edge. This effect may also occur when cells in tissue become smaller due to an increase of cell density through proliferation or compression; in other words, this means that the cells become smaller than the typical length scale of the chemical patterns that control cell migration. Then, one would not expect these chemical patterns to form (*Hubatsch et al., 2019*). Therefore, depending on the cell polarizability ($\Delta\epsilon$), there is an

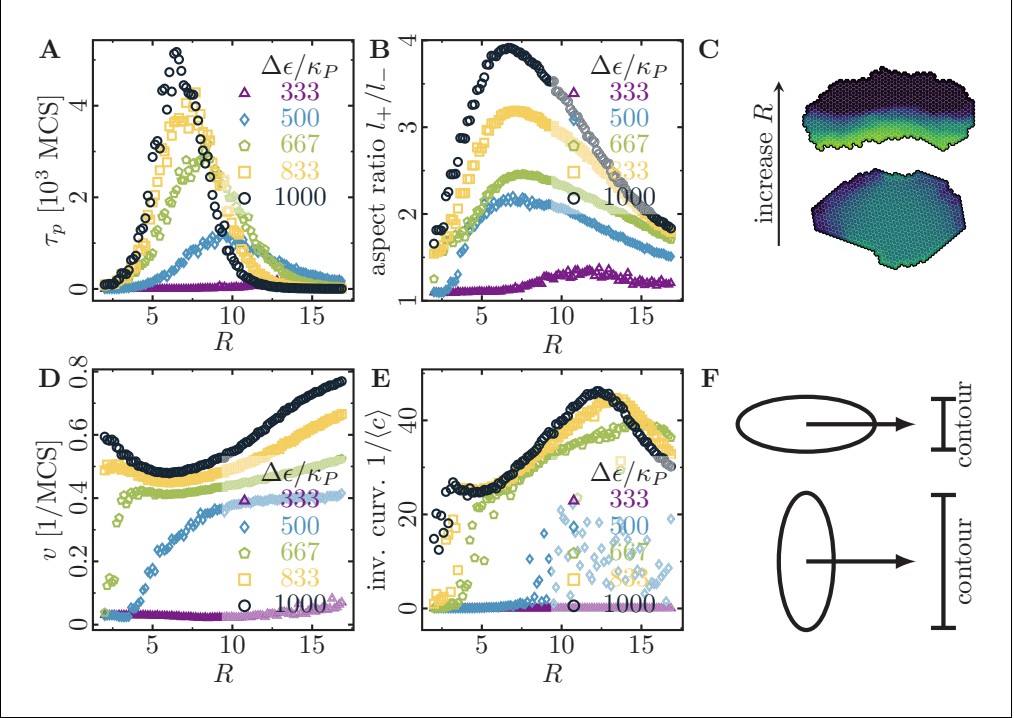

**Figure 3.** Migratory behavior of single cells as a function of the cell's signaling radius $R$ at different values for the maximal cytoskeletal polarity $\Delta\epsilon$. (Stiffness parameters $\kappa_P = 0.060$, $\kappa_A = 0.18$; average polarization field $\epsilon_0 = 225$; cell-substrate dissipation $D = 0$; cell-substrate adhesion penalty $\varphi = 0$; cytoskeletal update rate $\mu = 0.1$; 100 independent simulations for each set of parameters). (A) The persistence times of directed migration of single cells exhibit a pronounced maximum at an optimal signaling radius, which depends on cell polarizability. (B) The shapes of single cells exhibit a pronounced maximal elongation at an optimal signaling radius, which depends on cell polarizability. (C) The signaling radius critically determines the synchronicity of internal cytoskeletal remodeling processes. Small signaling radii frequently lead to transient formation of mutually independent lamellipodia at different positions around the cell perimeter, thereby interrupting persistent motion (reducing persistence times of directed migration). Large signaling radii lead to structurally stable front-rear polarization profiles across the entire cell body (long persistence times of directed migration). Color code: cell polarization; cf. color bar in **Figure 1C**. (D) The speed of single cells does not drop to zero even when their persistence time of directed migration vanishes. This indicates single cell rotations. (E) The inverse curvature of the cell trajectories as a function of the signaling radius. (F) Depending on whether a cell migrates along its long axis (top) or short axis (bottom), it has to move a different projected contour length. If each protrusion takes roughly the same amount of time, then migration along the long axis (top; cell has to move a smaller projected contour length) allows for greater cell speeds than migration along the short axis (bottom; cell has to move a larger projected contour length).

The online version of this article includes the following video for figure 3:

**Figure 3—video 1.** Single cell motility and shape for different signaling radii ($\Delta\epsilon = 60$, $\kappa_P = 0.060$).
https://elifesciences.org/articles/46842#fig3video1

optimal signaling radius that shows both maximal cell elongation and maximal cell persistence (**Figure 3A,B**).

Cells with low polarizability need a large signaling radius to feed the positive feedback mechanism and to form a single large cell front. In contrast, highly polarizable cells can already sustain the positive feedback mechanism with a short signaling radius and easily form at least one (or even multiple competing) short cell front(s). With increasing signaling radius, these cell fronts become increasingly correlated and finally merge. Surprisingly, at small signaling radii, we observed that highly polarizable cells slow down with increasing signaling radius (**Figure 3D**; yellow squares and black circles), in contrast to the behavior of cells with low polarizability. Furthermore, at large signaling radii, highly polarizable cells *speed up*, although their persistence time of directed migration has dropped to small values (cf. **Figure 3A,D**; blue diamonds, green pentagons, yellow squares and

black circles). To find an intuitive explanation for these observations, we inspected time-lapse videos of a cell at high polarizability ($\Delta\epsilon/\kappa_P = 1000$; cf. *Figure 3—video 1*, top row), which show a qualitative shift in cell behavior:

- For small signaling radii, $R = 2$, short polarization fronts 'pull' the cell behind them, allowing for transient polarization and quick but erratic movement along the long axis of the cell.
- For intermediate signaling radii, $R = 6$, broad and correlated polarization fronts emerge, and both the cell polarization and movement always orient themselves along the short axis of the cell.
- For large signaling radii $R = 15$, we observed circular motion of the cell; because of the large signaling radius, signals originating from the trailing edge affect the leading edge of the cell and vice versa. Due to this circular motion, the cell exhibits a non-zero speed and a vanishing persistence time of directed migration.

Therefore, we find that the cell can transiently polarize and migrate along its long axis for small signaling radii and for high polarizability. Furthermore, in a broad parameter regime, we find keratocyte-like motion and polarization along the short axis of the cell. Note that we do not consider the formation of stress fibers, which could lead to cell migration along the long axis in a broad parameter regime (*Kassianidou et al., 2019*). Such stress fibers could be modeled via a nematic field that represents the anisotropic part of the intracellular stress. Our counter-intuitive observation that cell migration along the long axis is faster than cell migration along the short axis can be explained as follows: If the cell migrates along its short axis, then it has to move a greater projected contour length than if it migrates along its long axis (*Figure 3F*). Considering that each protrusion takes roughly the same amount of time, migration along the long axis allows for greater cell speeds than migration along the short axis, because the cell has to spend less time to move a smaller projected contour length (*Figure 3F*).

To further characterize the single cell rotations that occur at large signaling radii, we determined the average curvature of the trajectories $\langle c \rangle = \langle \|\partial_s \hat{\mathbf{v}}(s) \cdot \hat{\mathbf{v}}(s)\| \rangle$, where $s$ is the contour length along the corresponding trajectory. Here, we averaged the tangent vector $\hat{\mathbf{v}}(s)$ over 10 Monte-Carlo steps to integrate out fluctuations that occur on short timescales (the internal dynamics of the cell has an intrinsic time scale of 10 Monte-Carlo steps due to our choice of the cytoskeletal update rate, $\mu = 0.1$). We find that the curvature of the trajectories has a pronounced minimum at large signaling radii (where the persistence time of directed migration vanishes), which indicates a transition from straight to circular trajectories (*Figure 3E*). Such a transition from persistent migration to single cell rotations was previously observed in experiments (*Lou et al., 2015*; *Raynaud et al., 2016*) and in theory (*Reeves et al., 2018*; *Allen et al., 2018*).

## Cell clusters on circular micropatterns

To assess the transition to collective cell motion, we next studied the dynamics of small cell groups confined to circular micropatterns (*Huang et al., 2005*; *Doxzen et al., 2013*; *Deforet et al., 2014*; *Segerer et al., 2015*). We implemented these structures in silico by setting $\varphi(\mathbf{x}) = 0$ inside a radius $r_0$ and $\varphi(\mathbf{x}) \to -\infty$ outside. During each simulation run, the number of cells was also kept constant by deactivating cell division. We previously employed this setup to compare our numerical results with actual experimental measurements, and found very good agreement (*Segerer et al., 2015*). Here, we generalize these studies and present a detailed analysis of the statistical properties of the collective dynamics of cell groups in terms of the key parameters of the computational model.

When adhesive groups of two or more motile cells are confined on a circular island, they arrange themselves in a state of spontaneous collective migration, which manifests itself in the form of coordinated and highly persistent cell rotations about the island's midpoint $\mathbf{x}_0$ (*Huang et al., 2005*; *Doxzen et al., 2013*; *Deforet et al., 2014*; *Segerer et al., 2015*). The statistics of these states of rotational motion provide insight into the influence of cellular properties on the group's ability to coordinate cell movements. To quantify collective rotations, we recorded the average signed angular velocity of the cell cluster $\omega(t) = \hat{\mathbf{e}}_z \cdot \langle \tilde{\mathbf{v}}(t) \times \tilde{\mathbf{R}}(t)/\|\tilde{\mathbf{R}}(t)\|^2 \rangle_\mathcal{C}$. Here, $\hat{\mathbf{e}}_z$ is the out-of-plane unit vector, $\langle \ldots \rangle_\mathcal{C}$ denotes an average with respect to the cell population, and $\tilde{\mathbf{v}}(t) = \mathbf{v}(t) - \langle \mathbf{v}(t) \rangle_\mathcal{C}$ as well as $\tilde{\mathbf{R}} = \mathbf{R}(t) - \langle \mathbf{R}(t) \rangle_\mathcal{C}$ measure the velocity and position of each cell relative to the cell cluster (we have omitted the indices that identify individual cells for the sake of convenience and clarity). The resulting random variables for the magnitude of the angular velocity of the cell assembly, $|\omega(t)|$, and the

average cell perimeter $P(t) \equiv \langle P_\alpha(t) \rangle_\mathcal{C}$ were then used to characterize the statistics of collective cell rotation. For each specific choice of simulation parameters, we monitored $|\omega(t)|$ and $P(t)$ for a set of 100 statistically independent systems, each of which was observed over $T_{\text{sim}} = 10^4$ MCS. From these data, we then computed the mean overall rotation speed $\langle |\omega| \rangle$, its standard deviation $\sigma_\omega$, and the standard deviation of the cell perimeter, $\sigma_P$.

*Figure 4* illustrates the characteristic properties of collective cell rotations in systems containing $|\mathcal{C}| = 4$ cells endowed with varying maximum cell polarity $\Delta\epsilon$ and varying cell contractility $\kappa_P$. Analogously to our observations for single cells, the statistical measures shown in *Figure 4A* do not separately depend on cell contractility and maximum cell polarity, but depend only on the *specific polarizability* $\Delta\epsilon/\kappa_P$. Overall, we find that upon increasing the specific polarizability there is a marked transition from a quiescent state to a state where the cells are collectively moving. Below a threshold value for the specific polarizability ($\Delta\epsilon/\kappa_P \approx 450$ in *Figure 4A*), the rotation speed $\langle |\omega| \rangle$ (purple curves in *Figure 4A*) vanishes and the cells are immobile. In this regime, which we term the *stagnation phase*, or $\mathcal{S}$-phase, cytoskeletal forces are too weak to initiate coherent cell rotation, and the system's dynamics is dominated by relatively strong contractile forces, which tend to arrest the system in a 'low energy' configuration. Beyond this threshold, we identify three distinct phases of collective cell rotation. In the $\mathcal{R}_1$-phase, we find a steep increase in the average rotation speed and a local maximum in the fluctuations of both cell shape and rotation speed; cf. green ($\sigma_P$) and blue ($\sigma_\omega$) curves in *Figure 4A*. Now, cytoskeletal forces are sufficiently large to establish actual membrane

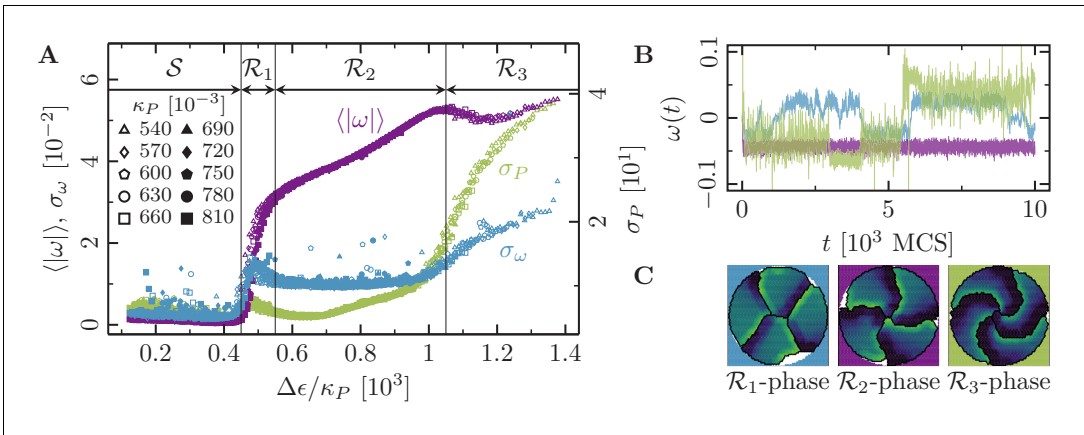

**Figure 4.** Phases of collective motion. (4-cell systems; confinement radius $r_0 = 30.6$; area stiffness $\kappa_A = 0.18$; average polarization field $\epsilon_0 = 225$; signaling radius $R = 5$; cytoskeletal update rate $\mu = 0.1$; cell-cell adhesion $B = 0$; cell-cell dissipation $\Delta B = 12$; cell-substrate dissipation $D = 0$; cell-substrate adhesion penalty $\varphi = 0$ ($r<r_0$), $\varphi \to -\infty$ ($r>r_0$); 100 independent simulations for each set of parameters). (A) Characteristic observables of collective cell rotation at different values of the cell perimeter stiffness parameter $\kappa_P$: mean ($\langle |\omega| \rangle$) and standard deviation ($\sigma_\omega$) of the magnitude of the cell cluster's angular velocity, and the standard deviation of the cell perimeter ($\sigma_P$). The statistics of collective cell motion depends only on the ratio of maximum cell polarity, $\Delta\epsilon$, to cell contractility, $\kappa_P$ (specific polarizability). (B) Representative angular trajectories and (C) cell shapes (color code represents cell polarization; cf. *Figure 1C*) for the different parameter regimes as described in the main text. The cellular dynamics in the different parameter regimes are shown in *Figure 4—video 1*, *Figure 4—video 2* and *Figure 4—video 3*.

The online version of this article includes the following video and figure supplement(s) for figure 4:

**Figure supplement 1.** Collective motion for varying number of cells at low polarizability.

**Figure supplement 2.** Collective motion for varying number of cells at intermediate polarizability.

**Figure supplement 3.** Collective motion for varying number of cells at high polarizability.

**Figure 4—video 1.** Collective rotations of 4 cells in the $\mathcal{R}_1$-phase ($\Delta\epsilon = 28$; $\Delta\epsilon/\kappa_P = 467$).

https://elifesciences.org/articles/46842#fig4video1

**Figure 4—video 2.** Collective rotations of 4 cells in the $\mathcal{R}_2$-phase ($\Delta\epsilon = 50$; $\Delta\epsilon/\kappa_P = 833$).

https://elifesciences.org/articles/46842#fig4video2

**Figure 4—video 3.** Collective rotations of 4 cells in the $\mathcal{R}_3$-phase ($\Delta\epsilon = 70$; $\Delta\epsilon/\kappa_P = 1167$).

https://elifesciences.org/articles/46842#fig4video3

protrusions against the contractile forces, and cells begin to rotate (*Figure 4B,C*). However, the contractile forces still dominate, such that cellular interfaces tend to straighten out and lamellipodium formation is sustained only over finite lifetimes. Thus, due to the dominance of contractile forces, the systems frequently experience transient episodes of stagnation and repeatedly change their direction of rotation (cf. blue trajectory in *Figure 4B*).

At intermediate values of specific polarizability ($\mathcal{R}_2$-phase), the cellular systems reach a regime of enduring rotational motion, where $\langle|\omega|\rangle$ varies linearly with the local specific polarizability, and where $\sigma_P$ and $\sigma_\omega$ exhibit a rather broad minimum (*Figure 4A*). In this regime, a range of 'optimal ratios' of cytoskeletal to contractile forces sustains stable cell shapes, and sets the stage for the formation of extended lamellipodia and the establishment of permanent front-rear polarizations of cells. As a result, the cells' persistence times of directed migration become very large, rendering cellular rotations strictly unidirectional within the observed time window (*Figure 4B*). Finally, at large values of the specific polarizability ($\mathcal{R}_3$-phase), the system's dynamics is dominated by cytoskeletal forces and the rotational speed $\langle|\omega|\rangle$ saturates at some maximal value. Due to the relatively small contractile forces, cell shapes tend to become unstable, as reflected in the growing variance of the cell perimeter $\sigma_P$ (green curve in *Figure 4A*). These instabilities in cell shape frequently lead to a loss of persistence in the rotational motion of the cells (growing $\sigma_\omega$; blue curve in *Figure 4A*).

## Tissue-level dynamics

As an application of our computational model at the tissue level, we considered a setup in which an epithelial cell sheet expands into free space. As in recent experimental studies (*Serra-Picamal et al., 2012*; *Sepúlveda et al., 2013*; *Trepat et al., 2009*; *Poujade et al., 2007*), we confined cells laterally between two fixed boundaries, within which they proliferated until they reached confluence; in the $y$-direction we imposed periodic boundary conditions. Then we removed the boundaries and studied how the cell sheet expands. In order to quantify tissue expansion, we monitored cell density and velocity, as well as the mechanical stresses driving the expansion process. *Figure 5* shows our results for two representative parameter regimes that highlight the difference between a dynamics dominated by cell motility in the absence of cell proliferation, and a contrasting regime where cells with low motility grow and divide depending on the local cell density. To simulate large numbers of cells, we decreased the amount of hexagons that are typically occupied by each cell (the simulation cost scales linearly with the summed area of all cells) by setting the average polarization field to $\epsilon_0 = 35$. For each set of parameters, we performed and averaged 100 independent simulations.

We first investigated how a densely packed pre-grown tissue of mitostatic cells with high polarizability (large $\Delta\epsilon$) expands into cell-free space upon removal of the confining boundaries at the tissue's lateral edges (*Figure 5A*). As the cells migrate into the cell-free space, we observe a strongly (spatially) heterogeneous decrease in the initially high and uniform cell density and mechanical pressure in the expanding monolayer (*Figure 5B,C*). This is quite distinct from the behavior of a homogeneous and ideally elastic thin sheet, which would simply show a homogeneous relaxation in density as it relaxes towards its rest state. Moreover, cell polarization and the ensuing active cell migration lead to inhomogeneously distributed traction stresses in the monolayer. After initial expansion of the monolayer, facilitated by high mechanical pressure, the cells at the monolayer edge begin to polarize outwards, which enhances outward front migration. These actively propagating cells exert traction on the trailing cells, and thereby yield a trailing region with negative stress (*Figure 5C*). Taken together, this gives rise to a characteristic X-shaped pattern in the kymograph of the total mechanical stresses $\langle\sigma_{xx}\rangle_y$ (*Figure 5C*). This profile closely resembles the first period of mechanical waves observed experimentally (*Serra-Picamal et al., 2012*). It illustrates how stress is transferred towards the center of the monolayer when cells are highly motile and collectively contribute to tissue expansion. At the end of the simulated time window, the cell density exhibited a minimum in the center of the sheet (*Figure 5B*). This is due to stretching of the central group of cells caused by the equally strong traction forces exerted by their migrating neighbors on both sides. Finally, the simulations also show that outward cell velocities increase approximately linearly with the distance from the center, confirming that in this configuration the entire cell sheet contributes to the monolayer expansion (*Figure 5D*).

To explore the possible range of tissue dynamics and expansion, we also investigated a qualitatively different parameter regime where cells are less densely packed and can also polarize less due

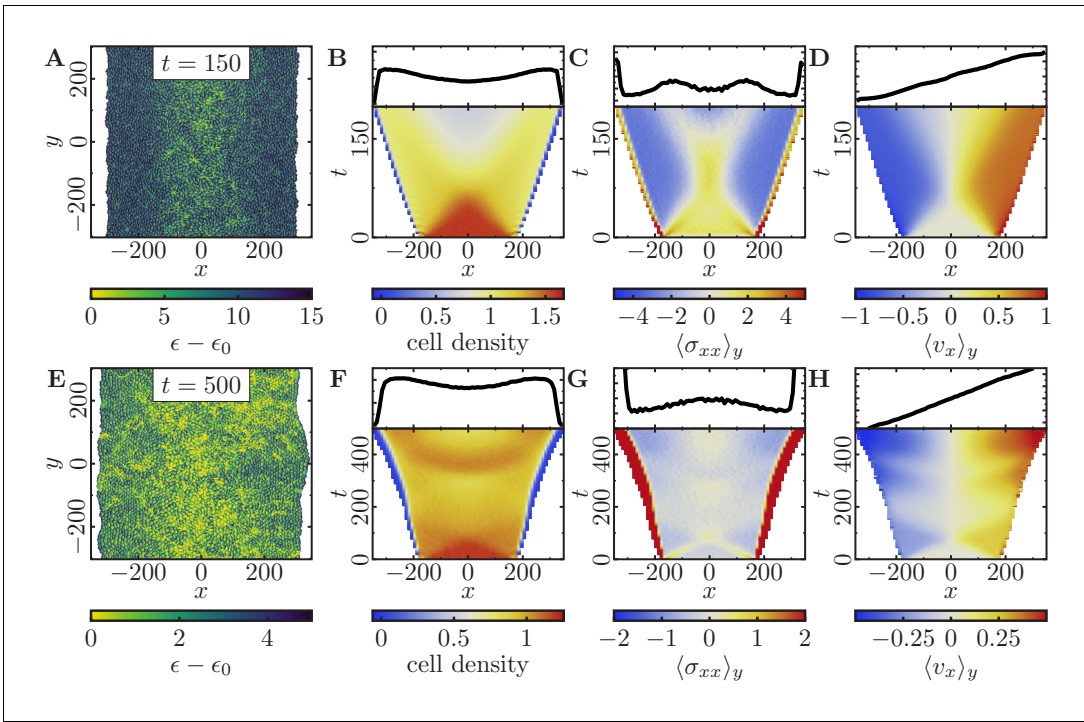

**Figure 5.** Expansion of a confluent epithelial cell sheet after removal of boundaries positioned at $x = \pm175$ for two different parameter settings. (Stiffness parameters $\kappa_P = 0.12$, $\kappa_A = 0.18$; average polarization field $\epsilon_0 = 35$; signaling radius $R = 2$; cytoskeletal update rate $\mu = 0.1$; cell-cell adhesion $B = 12$; cell-cell dissipation $\Delta B = 0$; cell-substrate dissipation $D = 0$; cell-substrate adhesion penalty $\varphi = 0$; 100 independent simulations for each set of parameters). (**A–D**) Tissue expansion for a migration-dominated setup without explicit cell growth and mitosis. (3300-cell system; maximum cell polarity $\Delta\epsilon = 30$). (**E–H**) Tissue expansion at low density and cell polarizability for a cell sheet comprised of dividing cells. (Initially a 2500-cell system; maximum cell polarity $\Delta\epsilon = 10$; growth time $T_g = 180$; division time $T_d = 20$; size threshold for cell growth $A_T = 1\,A_{ref}$, where $A_{ref}$ is the size of a solitary cell in equilibrium). (**A, E**) Snapshots of the polarization field $\epsilon$; cf. *Figure 5—video 1* and *Figure 5—video 2*. (**B, F**) Kymographs showing the cell density averaged over the $y$-direction and (*top*) final snapshots of the cell density profiles. (**C, G**) Kymographs showing the component $\sigma_{xx}$ of the stress tensor averaged over the $y$-direction and (*top*) snapshots of the stress profiles. (**D, H**) Kymographs showing the component $v_x$ of the cell velocities averaged over the $y$-direction and (*top*) final snapshot of the velocity profiles. The online version of this article includes the following video and figure supplement(s) for figure 5:

**Figure supplement 1.** Monolayer expansion depends on dissipation and cell polarizability.

**Figure 5—video 1.** Motility-dominated tissue dynamics.

https://elifesciences.org/articles/46842#fig5video1

**Figure 5—video 2.** Proliferation-dominated tissue dynamics.

https://elifesciences.org/articles/46842#fig5video2

to a narrower range of polarizability (*Figure 5E*). Here, the expansion of the monolayer is mainly driven by cell division, and cells keep dividing until they reach a homeostatic cell density (*Figure 5F*). Even though cells should typically exceed the threshold size and hence enter the growth phase at different times, we observe that the cell sheet exhibits periodic 'bursts' of growth (*Figure 5F*) coinciding with the total duration of a complete cell cycle (200 MCS) and alternating with cell migration (*Figure 5H*). These periodic 'bursts' can be explained as follows. Initially, the slightly compressed monolayer expands to relieve mechanical pressure. Due to this initial motion, the cells at the monolayer edge begin to polarize outwards. As in the previous case, where cell proliferation is absent (*Figure 5A–D*), the polarized cells enhance outward front migration and stretch the cells in the bulk of the cell sheet. For the same reasons as before, we observe a typical X-shaped stress pattern in the kymograph (*Figure 5G*), albeit less pronounced due to the lower polarizability of the cells (cf. *Figure 5C*). Because a broad region in the monolayer bulk is stretched by the actively migrating cell

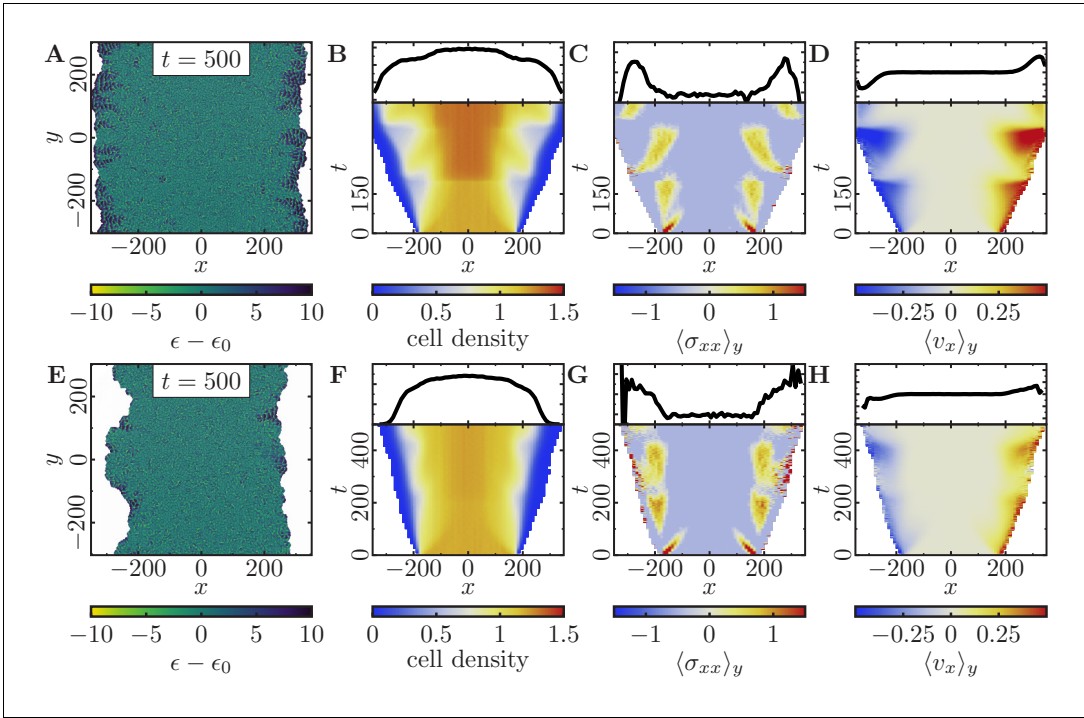

**Figure 6.** Expansion of a confluent epithelial cell sheet after removal of boundaries positioned at $x = \pm175$ for two different parameter settings that produce rough tissue fronts. (Initially a 2500-cell system; stiffness parameters $\kappa_P = 0.10$, $\kappa_A = 0.18$; average polarization field $\epsilon_0 = 35$; maximum cell polarity $\Delta\epsilon = 20$; signaling radius $R = 5$; cytoskeletal update rate $\mu = 0.1$; cell-cell adhesion $B = 5$; cell-cell dissipation $\Delta B = 10$; cell-substrate dissipation $D = 0$; cell-substrate adhesion penalty $\varphi = 0$; growth time $T_g = 180$; division time $T_d = 20$; 100 independent simulations for each set of parameters). (**A–D**) Tissue expansion at low density and cell polarizability for a cell sheet comprised of quickly dividing cells. (Size threshold for cell growth $A_T = 1.05\,A_{ref}$, where $A_{ref}$ is the size of a solitary cell in equilibrium). (**E–H**) Tissue expansion at low density and cell polarizability for a cell sheet comprised of slowly dividing cells. (Size threshold for cell growth $A_T = 1.10\,A_{ref}$, where $A_{ref}$ is the size of a solitary cell in equilibrium). (**A, E**) Snapshots of the polarization field $\epsilon$; cf. **Figure 6—video 1** and **Figure 6—video 2**. (**B, F**) Kymographs showing the cell density averaged over the $y$-direction and (*top*) final snapshots of the cell density profiles. (**C, G**) Kymographs showing the component $\sigma_{xx}$ of the stress tensor averaged over the $y$-direction and (*top*) final snapshots of the stress profiles. (**D, H**) Kymographs showing the component $v_x$ of the cell velocities averaged over the $y$-direction and (*top*) final snapshot of the velocity profiles.

The online version of this article includes the following video(s) for figure 6:

**Figure 6—video 1.** Weak monolayer roughening (fingering) in motility-dominated tissue with quick proliferation.
https://elifesciences.org/articles/46842#fig6video1

**Figure 6—video 2.** Strong monolayer roughening in motility-dominated tissue with slow proliferation.
https://elifesciences.org/articles/46842#fig6video2

fronts, these cells exceed the threshold size and begin growing approximately in phase. Once the mechanical pressure of the cell sheet is relieved, it will stop expanding (**Figure 5H**). However, cell growth and division once more lead to an increase in mechanical pressure (and cell density) in the monolayer (**Figure 5F,G**). This cycle of migration-dominated monolayer expansion and cell-density-dependent cell growth and division results in a periodic recurrence of the X-shaped stress pattern (**Figure 5G**), closely resembling the pattern observed in experiments (**Serra-Picamal et al., 2012**). On a sidenote, the synchronization of the cell division and cell migration phases by the deterministic portion of the cell cycle can be counteracted by introducing additional stochastic terms in the transition between the different phases of the cell cycle (cf. 'Cell proliferation and mitosis' in Appendix 1).

Note that the inhomogeneously distributed traction stresses in the monolayer, and its wave-like behavior, ultimately emerge from cell polarization and the ensuing active cell migration. Therefore,

these traction patterns would look much less prominent if one were to inhibit cell motility (compare *Figure 5C* with *Figure 5G*).

Finally, we investigated which parameters control the roughness of the tissue fronts. We found that increasing cell motility, or increasing cell-cell dissipation leads to rougher front morphologies (*Figure 5—figure supplement 1* and 'Velocity and roughness of spreading tissue' in Appendix 2). Therefore, we hypothesized that one could observe *fingering* of cell monolayers by adjusting the parameters accordingly:

- Increase of cell motility by decreasing the membrane stiffness and at the same time increasing polarizability and signaling radius of the cells.
- Increase of cell-cell dissipation and slight decrease of cell-cell adhesion.
- Slower and less homogeneously distributed cell division by increasing the cell threshold size.

Indeed, we then observe a drastic roughening of the cell fronts and small cohorts of cells that coherently move into cell-free space (*Figure 6*). This roughening is more pronounced if we further increase the threshold size that a cell has to exceed to initiate growth (cf. *Figure 6A,E*). Analogously to our previous discussion, we observe that an increasing mechanical pressure in the monolayer due to the division of cells initiates outward cell migration (*Figure 6B,F*). Then, cells in the tissue begin to polarize outwards and coordinate their motion with their neighboring cells, leading to small coordinated cell cohorts. As before, we also find distinct traction force patterns, as recurring waves of high stress travel backwards relative to the leading edges (*Figure 6C,G*), and distinct recurring velocity patterns (*Figure 6D,H*).

## Discussion

In this work, we have proposed a generalization of the cellular Potts model (*Graner and Glazier, 1992*). The model implements a coarse-grained routine that captures the salient features of cytoskeletal remodeling processes on subcellular scales, while being computationally tractable enough to allow for the simulation of entire tissues containing up to $\mathcal{O}(10^4)$ cells. We have used the model to study the transition from single-cell to cohort cell migration in terms of the interplay between the pertinent cellular functions. Specifically, we have demonstrated that our model consistently reproduces the dynamics and morphology of motile cells down to the level of solitary cells. Our studies also

**Table 1.** Source and parameter files used for each figure.
All source and parameter files are found in *Source data 1*.

| Figure | Simulation code | Processing code | Parameters |
|---|---|---|---|
| *Figure 2* | CPM_NoDivision | TrajectoryAnalysisSingle | single_Q |
| *Figure 2—figure supplement 1 (A-D)* | CPM_NoDivision | TrajectoryAnalysisSingle | single_DQ |
| *Figure 2—figure supplement 1 (E-H)* | CPM_NoDivision | TrajectoryAnalysisSingle | single_DM |
| *Figure 3* | CPM_NoDivision | TrajectoryAnalysisSingle | single_R |
| *Figure 4* | CPM_NoDivision | TrajectoryAnalysisCircularPattern | rotation_Q |
| *Figure 4—figure supplement 1* | CPM_NoDivision | TrajectoryAnalysisCircularPattern | rotation_N_R1 |
| *Figure 4—figure supplement 2* | CPM_NoDivision | TrajectoryAnalysisCircularPattern | rotation_N_R2 |
| *Figure 4—figure supplement 3* | CPM_NoDivision | TrajectoryAnalysisCircularPattern | rotation_N_R3 |
| *Figure 5 (A-D)* | CPM_Division | | wound_nodiv |
| *Figure 5 (E-H)* | CPM_Division | | wound_div |
| *Figure 5—figure supplement 1 (A-B)* | CPM_Division_Supplement | FrontAnalysis | wound_div_A |
| *Figure 5—figure supplement 1 (C-D)* | CPM_Division_Supplement | FrontAnalysis | wound_div_D |
| *Figure 5—figure supplement 1 (E, F)* | CPM_Division_Supplement | FrontAnalysis | wound_div_Q |
| *Figure 6 (A-D)* | CPM_Division | | wound_div_fing_1.0 |
| *Figure 6 (E-H)* | CPM_Division | | wound_div_fing_1.1 |
| *Appendix 2—figure 1* | CPM_NoDivision | TrajectoryAnalysisSingle | single_A |

reveal that cytoskeletal forces (relative to cell contractility), as well as the spatial organization of the cells' lamellipodia, significantly affect the statistics of cellular trajectories, both in the context of single-cell motion and in cohesive cell groups restricted to circular micropatterns. On larger scales, our simulation results suggest that the dynamics of expanding tissues strongly depends on the specific properties of the constituent cells. If monolayer expansion is driven by active cell migration throughout the tissue, then the cell sheet exhibits typical traction-force patterns and an X-shape in the corresponding kymograph. Additionally, a cell-density-dependent cell growth leads to a periodic recurrence of these traction-force patterns in a cycle of migration-dominated expansion and 'burst'-like cell proliferation.

Taken together, our results further highlight the intricacies of collective cell migration, which involves a multitude of intra- and inter-cellular signaling mechanisms operating at different scales in length and time. Establishing a comprehensive picture that incorporates and elucidates the mechanistic basis of these phenomena remains a pressing and challenging task. The multiscale modeling approach proposed here provides a direct link between subcellular processes and macroscopic dynamic observables, and might thus offer a viable route towards this goal.

## Materials and methods

The computational model is described in section 'Computational model'. The numerical implementation of the model is discussed in detail in Appendix 1. The parameter files and source files associated with the figures are given in *Table 1*.

## Acknowledgements

FT, AG, MR and EF designed research, performed research, and wrote the paper. AG acknowledges support by a DFG fellowship through the Graduate School of Quantitative Biosciences Munich (QBM). EF acknowledges support by the German Excellence Initiative via the program 'NanoSystems Initiative Munich' (NIM) and by the Deutsche Forschungsgemeinschaft (DFG) via Collaborative Research Center (SFB) 1032 (project B02). We thank Felix Kempf, Felix Segerer, Sophia Schaffer and Joachim Rädler for stimulating discussions.

## Additional information

### Funding

| Funder | Grant reference number | Author |
| --- | --- | --- |
| German Excellence Initiative | NanoSystems Initiative Munich (NIM) | Erwin Frey |
| Deutsche Forschungsgemeinschaft | Collaborative Research Center (SFB) 1032 (project B02) | Erwin Frey |
| Deutsche Forschungsgemeinschaft | Graduate School of Quantitative Biosciences Munich (QBM) | Andriy Goychuk |

The funders had no role in study design, data collection and interpretation, or the decision to submit the work for publication.

### Author contributions

Florian Thüroff, Conceptualization, Software, Formal analysis, Validation, Investigation, Visualization, Methodology; Andriy Goychuk, Conceptualization, Data curation, Software, Formal analysis, Validation, Investigation, Visualization, Methodology; Matthias Reiter, Software, Formal analysis, Validation, Investigation, Visualization, Methodology; Erwin Frey, Conceptualization, Resources, Formal analysis, Supervision, Funding acquisition, Validation, Investigation, Methodology, Project administration

#### Author ORCIDs
Andriy Goychuk (iD) https://orcid.org/0000-0001-6776-9437
Erwin Frey (iD) https://orcid.org/0000-0001-8792-3358

#### Decision letter and Author response
Decision letter https://doi.org/10.7554/eLife.46842.sa1
Author response https://doi.org/10.7554/eLife.46842.sa2

## Additional files

### Supplementary files
• Source code 1. Simulation code, processing code and parameter files.

• Transparent reporting form

### Data availability
We have uploaded the source code used in the main part of our study as well as the one used in the appendix. Furthermore, we have provided the full list of parameters in the figure captions, as well as exemplary parameter files for all applicable figures.

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

## Appendix 1

# Computational model

In this section, we describe in detail the implementation of our computational model, which has been outlined briefly in the main text. While the biological rationale behind our modeling approach has been discussed in the main text, our focus here is on the technical aspects and the details of the numerical implementation. To facilitate subsequent discussions on implementation details, we start by introducing some model-specific terminology which will be used throughout this section to illustrate the mechanics of our model.

# Computational grid

The basic data structure that underlies our computational model is referred to as the *grid*; see *Appendix 1—figure 1*. The grid itself is implemented as a regular, space-filling lattice with lattice vectors $\{\mathbf{x}_i\}_{i=1,\dots,N}$. Each lattice vector $\mathbf{x}_i$ is understood to represent its associated Voronoi cell which will be referred to as *grid site*. To be specific, we consider triangular tilings $\{\mathbf{x}_i\}_{i=1,\dots,N}$, such that each grid site is a hexagon, which is surrounded by 6 nearest-neighbor sites that define the neighborhood $\mathcal{N}_k$ of $\mathbf{x}_k$:

$$\mathcal{N}_k = \left\{ \mathbf{x}_j \mid \mathbf{x}_j \text{ is nearest neighbor of } \mathbf{x}_k \right\} \tag{S1}$$

Overall, the grid represents our general notion of (discretized) space, and each grid site holds information specific to cells as well as to environmental factors. In what follows, distances on this spatial grid will be measured in units of the distance between the midpoints of neighboring lattice sites, i.e.

$$\|\mathbf{x}_k - \mathbf{x}_j\| = 1 \iff j \in \mathcal{N}_k. \tag{S2}$$

This then implies for the side length $\ell$ and the two-dimensional volume (area) $a$ of each hexagonal grid site: $\ell = 1/\sqrt{3}$ and $a = 3\sqrt{3}\,\ell^2/2$.

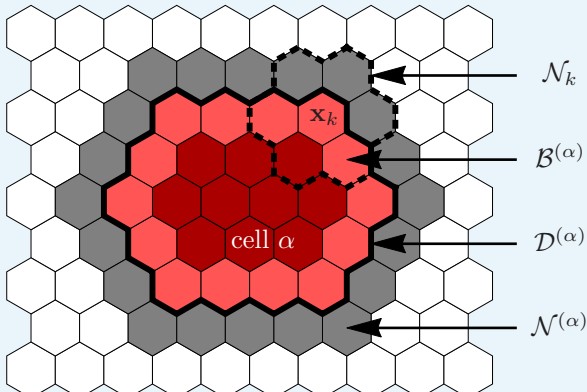

**Appendix 1—figure 1.** Illustration of the various sets defining a cell and its environment. Grid sites occupied by cell $\alpha$, i.e. its domain $\mathcal{D}^{(\alpha)}$, are indicated in red colors. The cell's membrane sites, $\mathcal{B}^{(\alpha)}$, are indicated by the lighter red color, the cell's immediate neighborhood, $\mathcal{N}^{(\alpha)}$, is indicated in gray. Elementary events involving cell $\alpha$ always involve one grid site in $\mathcal{B}^{(\alpha)}$ and one grid site in $\mathcal{N}^{(\alpha)}$. For the hexagonal lattices used in this work, each grid site $\mathbf{x}_k$ is surrounded by 6 nearest neighbors which we collectively denote by $\mathcal{N}_k$.

## Representation of biological cells

In the spirit of the cellular Potts model (*Graner and Glazier, 1992*; *Glazier and Graner, 1993*), each cell is represented by a simply connected set of lattice sites

$$\mathcal{D}^{(\alpha)} = \left\{ \mathbf{x}_k \mid c(\mathbf{x}_k) = \alpha \right\}, \tag{S3}$$

where the indicator function $c(\mathbf{x}_k)$ gives the index of the cell occupying $\mathbf{x}_k$. Here and in the following, we use latin indices to reference lattice sites, and greek indices to reference cells. The set $\mathcal{D}^{(\alpha)}$ used to represent the spatial extension of cell $\alpha$, will be referred to as the *domain* of cell $\alpha$. In our model, each grid site $\mathbf{x}_k$ can be occupied by at most one cell (i.e. we do not allow for overlapping cell domains). The absence of cells at $\mathbf{x}_k$ is numerically implemented by negative values of the indicator function, $c(\mathbf{x}_k) < 0$. Following this terminology, the area and the perimeter of cell $\alpha$ are given by:

$$A_\alpha = a \sum_{k=1}^{N} \delta_{\alpha, c(\mathbf{x}_k)} = \frac{3\sqrt{3}}{2} \ell^2 \sum_{k=1}^{N} \delta_{\alpha, c(\mathbf{x}_k)}, \tag{S4a}$$

$$P_\alpha = \ell \sum_{k=1}^{N} \sum_{\mathbf{x}_l \in \mathcal{N}_k} \delta_{\alpha, c(\mathbf{x}_k)} (1 - \delta_{\alpha, c(\mathbf{x}_l)}). \tag{S4b}$$

## Model dynamics

### Protrusion and retraction of cells

Biological cells are highly dynamic entities which constantly change shape and move around in space. To reflect this dynamic behavior computationally, the domain $\mathcal{D}^{(\alpha)}$ of cell $\alpha$ changes over time. The evolution of cell shape and position, as represented by $\mathcal{D}^{(\alpha)}$, proceeds via a succession of *elementary events*. In our numerical model, elementary events come in one of two basic flavors: *protrusion events* and *retraction events*. During a protrusion event, cell $\alpha$ (referred to as *source cell*) incorporates one grid site $\mathbf{x}_t$ (referred to as *target grid site*) from its neighborhood $\mathcal{N}^{(\alpha)}$,

$$\mathcal{D}^{(\alpha)}_{\text{old}} \to \mathcal{D}^{(\alpha)}_{\text{new}} = \mathcal{D}^{(\alpha)}_{\text{old}} \cup \{\mathbf{x}_t\}, \quad \mathbf{x}_t \in \mathcal{N}^{(\alpha)}, \tag{S5}$$

thereby increasing its cellular domain by one grid site. Here, the neighborhood of cell $\alpha$, $\mathcal{N}^{(\alpha)}$, is defined as

$$\mathcal{N}^{(\alpha)} = \left\{ \mathbf{x}_l \mid \min_{\mathbf{x}_k \in \mathcal{D}^{(\alpha)}} \|\mathbf{x}_l - \mathbf{x}_k\| = 1 \right\}. \tag{S6}$$

During a retraction event, source cell $\alpha$ expels one of its membrane grid sites $\mathbf{x}_s \in \mathcal{B}^{(\alpha)}$,

$$\mathcal{D}^{(\alpha)}_{\text{old}} \to \mathcal{D}^{(\alpha)}_{\text{new}} = \mathcal{D}^{(\alpha)}_{\text{old}} \setminus \{\mathbf{x}_s\}, \quad \mathbf{x}_s \in \mathcal{B}^{(\alpha)}_{\text{old}}, \tag{S7}$$

where the set of membrane grid sites $\mathcal{B}^{(\alpha)}$ is defined as

$$\mathcal{B}^{(\alpha)} = \left\{ \mathbf{x}_k \in \mathcal{D}^{(\alpha)} \mid \min_{\mathbf{x}_l \in \mathcal{N}^{(\alpha)}} \|\mathbf{x}_k - \mathbf{x}_l\| = 1 \right\}. \tag{S8}$$

Protrusion and retraction events are the numerical analogs of cell protrusions and cell retractions.

In implementing the reassignment rules, *Equation S5* and *Equation S7*, we have to take into account that cellular domains must not overlap. For solitary cells moving in free space this does not imply any restrictions, and *Equation S5* and *Equation S7* apply directly. In the bulk of a confluent monolayer of adhesive cells, however, any protrusion of source cell $\alpha$ into the domain of cell $\beta$ (referred to as *target cell*) must be accompanied by a corresponding

retraction event $\mathcal{D}_{\mathrm{old}}^{(\beta)} \to \mathcal{D}_{\mathrm{new}}^{(\beta)} = \mathcal{D}_{\mathrm{old}}^{(\beta)} \setminus \{\mathbf{x}_t\}$, where $\mathbf{x}_t$ denotes the target grid site annexed by cell $\alpha$. We emphasize, however, that the reverse is not generally true. If source cell $\alpha$ retracts, i.e. loses one of its boundary grid sites $\mathbf{x}_s \in \mathcal{B}^{(\alpha)}$, the lost grid site $\mathbf{x}_s$ faces either one of two conceivable fates: If, on the one hand, cohesion among cells is sufficiently strong (cf. section 'Rupture of cell contacts' for a definition of the notion 'sufficiently strong'), then the retraction of cell $\alpha$ exerts a pulling force on one of its neighboring cells $\beta$ (the target cell) and forces the target cell to fill the emerging void at $\mathbf{x}_s$, i.e. $\mathcal{D}_{\mathrm{old}}^{(\beta)} \to \mathcal{D}_{\mathrm{new}}^{(\beta)} = \mathcal{D}_{\mathrm{old}}^{(\beta)} \cup \{\mathbf{x}_s\}$, where $\mathbf{x}_s$ denotes the grid site lost by cell $\alpha$. On the other hand, if adhesion between cells is weak, then retraction of the source cell $\alpha$ can lead to a rupture of pre-existing cell contacts between $\alpha$ and other cells at $\mathbf{x}_s$, such that the lost grid site $\mathbf{x}_s$ becomes free space [$c(\mathbf{x}_s) = \alpha \geq 0 \to c(\mathbf{x}_s) < 0$]. Details on the actual implementation of cell rupture are discussed in section 'Rupture of cell contacts'.

## Monte-Carlo scheme

In the spirit of a standard Monte-Carlo scheme, the actual simulation proceeds via a succession of *Monte-Carlo steps*, where each Monte-Carlo step (MCS) propagates the state of the simulated cell population from time $t$ to time $t + \Delta t$, where we set the time step to $\Delta t \equiv 1$. One MCS consists in a series of attempts to perform elementary events, originating from randomly chosen membrane grid sites of randomly chosen cells. The duration of one MCS, i.e. the actual number of attempted elementary events, is chosen such that each of the cells' membrane segments is given the opportunity to attempt, on average, one elementary event per MCS. During each MCS, cell domains $\mathcal{D}^{(\alpha)}$ as well as the numerical values of cell areas $A_\alpha$ and perimeters $P_\alpha$ are updated 'on the fly', while the cells' polarization fields are updated only once at the end of each MCS; cf. section 'Cytoskeletal structures and focal adhesion' for the details of this update rule. The simulation then proceeds along the following Monte-Carlo scheme:

1. Initialize the cell population and define the duration of the simulation, i.e. the number of MCS, $T_{\mathrm{sim}}$, to be performed.
2. Set the simulation time $t = 0$.
3. Perform the next MCS; this step is further detailed below.
4. Update polarization fields (cf. section 'Cytoskeletal structures and focal adhesion').
5. Set $t = t + \Delta t$, where $\Delta t \equiv 1$.
6. Repeat steps 3–5 while $t < T_{\mathrm{sim}}$.

The implementation of a MCS, i.e. the sequence of elementary events, is based on the following general considerations:

i. *Choice of source and target grid sites.* Each elementary event $\mathcal{T}$ originates from a membrane grid site $\mathbf{x}_s \in \mathcal{B}^{(\alpha)}$ of some cell $\alpha$, referred to as *source cell*. This membrane grid site will be referred to as *source grid site*. In addition, each elementary event involves a second grid site which lies in the neighborhood of the source grid site $\mathbf{x}_s$ and which is not currently occupied by cell $\alpha$: $\mathbf{x}_t \in \mathcal{N}_s \setminus \mathcal{D}^{(\alpha)}$. In what follows, this additional grid site $\mathbf{x}_t$ will be referred to as *target grid site*. This grid site may either be an empty substrate site or a membrane site of another cell $\beta$, in which case the respective cell will be referred to as *target cell*. While the source grid site determines the location of the attempted elementary event, the target grid site determines the direction along which the elementary event is bound to proceed.

ii. *Monte-Carlo method to generate the system's dynamics.* As mentioned above, the actual dynamics of cells in our computational model is driven by a succession of elementary events, whose cumulative effects over time allow cells to change shapes and to move relative to the substrate as well as relative to each other. Following a standard Monte-Carlo procedure, the probability of occurrence of elementary events $\mathcal{T}$ is determined by a goal function $p(\mathcal{T})$ [cf. point (iii) below]. However, since elementary events come in two basic flavors, protrusions $\mathcal{T}_{\mathrm{pro}}$ and retractions $\mathcal{T}_{\mathrm{ret}}$, their actual occurrence is controlled by a two-step process, once source and target grid sites have been determined: In a first step, two alternative scenarios are proposed where either the source cell protrudes toward $\mathbf{x}_t$, or retracts from $\mathbf{x}_s$. Then, a decision is made with equal probabilities as to whether one attempts $\mathcal{T}_{\mathrm{pro}}$ or $\mathcal{T}_{\mathrm{ret}}$. In a second step, the goal function $p$ is used to compute the

occurrence probability of the attempted event $\mathcal{T}$. Finally, this elementary event $\mathcal{T}$ is being accepted with probability $p(\mathcal{T})$.

iii. *Choice of the goal function $p(\mathcal{T})$.* As has been detailed above, we use a goal function $p(\mathcal{T})$ to control the occurrence and acceptance of elementary events $\mathcal{T}$. Following the standard cellular Potts model (**Graner and Glazier, 1992**; **Glazier and Graner, 1993**), this goal function takes into account the effects of cell contractility and cell-cell adhesion, using, however, a slightly different implementation; cf. sections 'Cell contractility' and 'Cell adhesion'. In addition, we generalized the definition of the goal function $p(\mathcal{T})$ to explicitly take into account a simplified model of cytoskeletal structures and the ensuing polarization of cells. The actual definition of the goal function will be developed in section 'Implementation of cellular traits', where, moreover, details concerning the implementation of the cell polarization model will be discussed.

The implementation of a single MCS loop is then given by the following simulation scheme:

1. Determine the current number of trials per Monte-Carlo step (MCS), $K = \sum_\alpha P_\alpha / \ell$, and set the trial counter $n = 0$.
2. With equal probability, choose a cell membrane segment (cf. solid black line in **Appendix 1—figure 1**) from a random cell $\alpha$ of the cell population. Because the cell membrane represents the border between lattice sites occupied by cell $\alpha$ and unoccupied by cell $\alpha$, the chosen membrane segment automatically defines the source grid site $\mathbf{x}_s \in \mathcal{B}^{(\alpha)}$ and the corresponding target grid site $\mathbf{x}_t \in \mathcal{N}^{(\alpha)} \cap \mathcal{N}_s$.
3. With equal probability, choose whether to attempt a protrusion event ($\mathcal{T}_{\mathrm{pro}}$) or a retraction event ($\mathcal{T}_{\mathrm{ret}}$).
4. Compute the prospective acceptance probability $p(\mathcal{T}_{\mathrm{pro/ret}})$ corresponding to the attempted event, and decide whether to accept the attempted event on the basis of this probability.
5. If the attempted elementary event has been accepted, then update the cellular domains of source cell $\alpha$ and opponent cell $\beta$; for details see section 'Cell domain update routine'.
6. If $n < K$, set $n \to n + 1$ and then repeat steps 2 through 5.

## Implementation of cellular traits

In this section, we discuss the various contributions of cellular traits to the overall acceptance probability $p(\mathcal{T})$ of an elementary event $\mathcal{T}$. Specifically, our model takes into account cell contractility, the assembly and disassembly of cytoskeletal structures, cell-cell adhesion, and focal adhesions. We will assume that each of these cellular properties contributes independently to the acceptance probability $p$, such that

$$p = \min\{1, \, p_{\mathrm{cont}} \cdot p_{\mathrm{cyto}} \cdot p_{\mathrm{adh}}\}. \tag{S9}$$

Anticipating our discussions in section 'Cytoskeletal structures and focal adhesion', the effects due to focal adhesions have been combined with the effects due to assembly and disassembly of cytoskeletal structures in $p_{\mathrm{cyto}}(\mathcal{T})$. In the following sections, we give detailed discussions for each of these contributions, separately.

### Cell contractility

In biological cells, membrane fluctuations are constrained by elastic forces and contractile cytoskeletal structures, which play a vital role in cell migration (**Alberts et al., 2015**; **Raucher and Sheetz, 2000**; **Friedl, 2004**). In our computational approach, we take cell contractility into account by assigning a contractile 'energy'

$$\mathcal{H}_{\mathrm{cont}} = \sum_\alpha \left[ \kappa_P^{(\alpha)} P_\alpha^2 + \kappa_A^{(\alpha)} A_\alpha^2 \right], \tag{S10}$$

with positive coupling constants $\kappa_P^{(\alpha)}$ and $\kappa_A^{(\alpha)}$ characterizing the contractility of cell $\alpha$; for empty substrate sites ($\alpha < 0$) we set $\kappa_P^{(\alpha)} = \kappa_A^{(\alpha)} = 0$. According to **Equation S10**, the cell's 'contractile energy' increases with increasing cell perimeter and increasing cell area. The model Hamiltonian $\mathcal{H}_{\mathrm{cont}}$ can then be used to specify the contractile contribution to the goal function $p(\mathcal{T})$. To this end, let $\Delta\mathcal{H}_{\mathrm{cont}}(\mathcal{T})$ denote the contractile contribution to the energy

difference entailed by accepting an elementary event $\mathcal{T}$. Following a standard Metropolis algorithm, we then define

$$p_{\mathrm{cont}}(\mathcal{T}) := \exp[-\Delta\mathcal{H}_{\mathrm{cont}}(\mathcal{T})/k_{\mathrm{B}}T], \tag{S11}$$

where we set the effective thermal energy to $k_{\mathrm{B}}T \equiv 1$. The contractile 'energy', *Equation S10*, is similar to the corresponding energy model commonly used in cellular Potts models (*Ouchi et al., 2003*). Unlike the standard cellular Potts model, however, where a target area and target perimeter are used to keep the simulated cells from collapsing, the energetic contribution in *Equation S10* strictly contracts the cell's body. As will be detailed in the next section, to counteract these contractile forces, we explicitly model cytoskeletal structures within each cell, which provide outward pushing forces to balance cell contraction.

## Cytoskeletal structures and focal adhesion

The cytoskeleton plays key roles both in maintaining the mechanical integrity of the cell and in the process of active cell migration (*Alberts et al., 2015*; *Friedl, 2004*; *Mogilner, 2009*). Our model design aims at achieving high computational efficiency to allow for the simulation of very large cell numbers (currently, cell numbers up to $\mathcal{O}(10^4)$ can be achieved at acceptable computation times) and, at the same time, to capture the essential effects of cytoskeletal dynamics to attain meaningful results down to the level of single cells. Thus, instead of accounting for a detailed biochemical description by means of reaction-diffusion networks (*Marée et al., 2006*; *Marée et al., 2012*), we resort to a simplified implementation of the most pertinent features of cytoskeletal dynamics. Specifically, we propose a rule-based algorithm to model cytoskeletal structures and to assess the integrated effects of cell polarity, cell contractility and adhesion on the collective dynamics of cells as parts of larger groups.

To this end, we define a scalar field $\epsilon(\mathbf{x}_n)$, $\mathbf{x}_n \in \mathcal{D}^{(\alpha)}$, on the domain of each cell $\alpha$. The local quantity $\epsilon(\mathbf{x}_n)$ will be referred to as *polarization field* and is taken to be a measure for the density of cytoskeletal structures at position $\mathbf{x}_n$ within the cell's body. The field variable $\epsilon(\mathbf{x}_n)$ is dynamically updated as the simulation progresses, reflecting cytoskeletal remodeling. To set up a system of rules underlying the actual implementation of these cytoskeletal remodeling processes, we resort to the following biologically motivated premises:

1. *The scalar polarization field $\epsilon$ is bounded*: The dynamics of cytoskeletal remodeling not only depends on the local number (density) of actin monomers and polymers, but also on a multitude of accessory proteins controlling cytoskeleton assembly and disassembly. Several biological factors—including the action of sequestering proteins like thymosin-β4, which act to suppress actin polymerization, limited amounts of nucleating proteins like the activated Arp2/3 complex, and the action of capping proteins—keep the local density of actin filaments bounded. We, therefore, introduce bounds for the *polarization field*: $\epsilon(\mathbf{x}_n, t) \in [\epsilon_0 - \Delta\epsilon/2, \; \epsilon_0 + \Delta\epsilon/2]$. These bounds are cell-type specific. While the upper bound $\epsilon_0 + \Delta\epsilon/2$ mainly reflects the limited availability of protein resources, the lower bound $\epsilon_0 - \Delta\epsilon/2$ serves to prevent cells from collapsing.

2. *Regulatory proteins affect assembly and disassembly of cytoskeletal structures:* The assembly and disassembly of cytoskeletal structures, numerically encoded by $\epsilon(\mathbf{x}_n)$, is regulated by a myriad of accessory proteins. In our computational model we simplify these complex processes by resorting to a single 'bookkeeping variable' which we will refer to as 'regulatory factors'. Its local level is stored as an integer variable $F(\mathbf{x}_n)$ for each grid site $\mathbf{x}_n \in \mathcal{D}^{(\alpha)}$. We use $F(\mathbf{x}_n)$ to implement the overall action of regulatory cytoskeletal proteins in an effective and collective manner. Specifically, since the formation of lamellipodial structures depends on active nucleation promoting factors (*Pollard and Borisy, 2003*), we assume that positive levels, $F(\mathbf{x}_n) > 0$, reflect local conditions in support of network-assembly, whereas negative levels, $F(\mathbf{x}_n) < 0$, represent predominantly degrading (or disassembly) conditions. For neutral levels, $F(\mathbf{x}_n) = 0$, the network gradually restores its rest state.

3. *Feedback between cytoskeletal structures and regulatory factors:* The activities of accessory cytoskeletal proteins which regulate the local levels of cytoskeletal structures are themselves controlled by a number of mechanical and chemical signals received by the cell. Here and in the following, our focus will be on mechanical signals. For example, important regulatory proteins like the Arp2/3 complex are activated locally at the cell membrane, from where they

diffuse into the bulk of the cell until they are bound by actin (**Kovacs et al., 2002**; **Pollard and Borisy, 2003**; **Leckband et al., 2011**). Adopting a coarse level of description, this diffusion-degradation dynamics entails a finite range of regulatory proteins, which are activated at the cell's membrane. In our model, we use the integer variable $F(\mathbf{x}_n)$ to implement this propagation of mechanical information, perceived by cell $\alpha$ at its periphery $\mathcal{B}^{(\alpha)}$, across a certain spatial distance $R$. The local levels of $F(\mathbf{x}_n)$ are continuously updated as the MCS loop progresses. The actual update procedure is given by the following set of rules; cf. **Appendix 1—figure 2**:

- if a *protrusion event* has been accepted at the source site $\mathbf{x}_s \in \mathcal{B}^{(\alpha)}$ (source cell: $\alpha$; target cell: $\beta$), then for all sites $\mathbf{x}_n$ within a range $R$ (i.e. $\|\mathbf{x}_n - \mathbf{x}_s\| < R$) the integer variable signifying regulatory factors is incremented up and down for the protruding and the retracting cell, respectively:

$$F(\mathbf{x}_n) \rightarrow \begin{cases} F(\mathbf{x}_n) + 1, & \mathbf{x}_n \in \mathcal{D}^{(\alpha)}, \\ F(\mathbf{x}_n) - 1, & \mathbf{x}_n \in \mathcal{D}^{(\beta)}. \end{cases} \tag{S12a}$$

- Similarly, if a *retraction event* has been accepted at the source site $\mathbf{x}_s \in \mathcal{B}^{(\alpha)}$, and the (local) cell contact between source cell $\alpha$ and target cell $\beta$ has remained intact, then within a range $R$ one applies the inverse update rule:

$$F(\mathbf{x}_n) \rightarrow \begin{cases} F(\mathbf{x}_n) - 1, & \mathbf{x}_n \in \mathcal{D}^{(\alpha)}, \\ F(\mathbf{x}_n) + 1, & \mathbf{x}_n \in \mathcal{D}^{(\beta)}. \end{cases} \tag{S12b}$$

- If a *retraction event* has been accepted at the source site $\mathbf{x}_s \in \mathcal{B}^{(\alpha)}$, and in addition the (local) cell contact between source cell $\alpha$ and target cell $\beta$ has *ruptured*, then the regulatory factors are reduced only within a range $R$ in the retracting cell:

$$F(\mathbf{x}_n) \rightarrow \begin{cases} F(\mathbf{x}_n) - 1, & \mathbf{x}_n \in \mathcal{D}^{(\alpha)}, \\ \text{no update}, & \text{else}. \end{cases} \tag{S12c}$$

Finally, if the target grid site $\mathbf{x}_t$ is not occupied by any cell (substrate is indicated by $\beta < 0$) prior to the elementary event, then only the first two lines in the above update scheme apply.

By virtue of the above update scheme, **Equation S12**, 'regulatory factors' are continuously distributed across each cell's domain $\mathcal{D}^{(\alpha)}$ as the current MCS progresses. At the end of each MCS, the accumulated (local) values of $F(\mathbf{x}_n)$ are used to update the local values of the polarization field $\epsilon(\mathbf{x}_n)$ inside each cell $\alpha \geq 0$ ($\mathbf{x}_n \in \mathcal{D}^{(\alpha)}$): We assume that for positive values, $F(\mathbf{x}_n) > 0$, there is assembly of cytoskeletal structures and $\epsilon$ is increased by an amount proportional to the distance of $\epsilon$ from its upper bound $\epsilon_0 + \Delta\epsilon/2$:

$$\epsilon(\mathbf{x}_n, t + \Delta t) = \epsilon(\mathbf{x}_n, t) + \Delta t \, \mu \left[ \epsilon_0 + \Delta\epsilon/2 - \epsilon(\mathbf{x}_n, t) \right], \tag{S13a}$$

where the time step is defined as $\Delta t \equiv 1$. Thereby $\epsilon_0 + \Delta\epsilon/2$ is a fixed point of this map and limits the build-up of cytoskeletal structures. In contrast, for negative values, $F(\mathbf{x}_n) < 0$, disassembly prevails, and we assume that $\epsilon$ then tends towards its lower bound $\epsilon_0 - \Delta\epsilon/2$:

$$\epsilon(\mathbf{x}_n, t + \Delta t) = \epsilon(\mathbf{x}_n, t) + \Delta t \, \mu \left[ \epsilon_0 - \Delta\epsilon/2 - \epsilon(\mathbf{x}_n, t) \right], \tag{S13b}$$

where the time step is defined as $\Delta t \equiv 1$. Neutral values, $F(\mathbf{x}_n, t) = 0$, lead to relaxation of $\epsilon$ towards a resting state

$$\epsilon(\mathbf{x}_n, t + \Delta t) = \epsilon(\mathbf{x}_n, t) + \Delta t \, \mu \left[ \epsilon_0 - \epsilon(\mathbf{x}_n, t) \right], \tag{S13c}$$

where the time step is defined as $\Delta t \equiv 1$. The parameter $\mu$ signifies the rate at which cytoskeletal structures respond to the regulatory factors $F$. For the parameters and cell sizes used in this work ($\epsilon_0 = \mathcal{O}(100)$ and $\Delta\epsilon = \mathcal{O}(10)$, and each cell occupying approximately 1000 grid sites) we set $\mu = 0.1$.

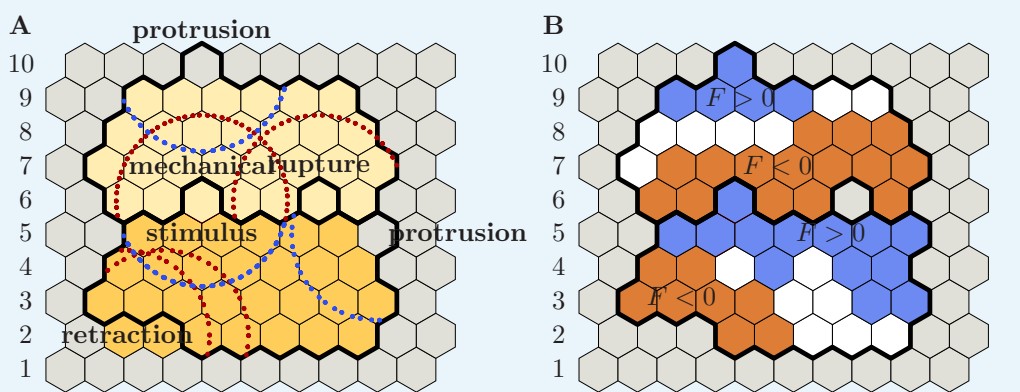

**Appendix 1—figure 2.** Distribution of regulatory factors on the basis of accepted elementary events. For ease of reference, grid rows have been numbered from 1 to 10. *Left* (**A**): Solid black lines indicate cells' membrane positions *after* acceptance of the respective elementary event; colors indicate cellular domains *before* the respective elementary event has been accepted (gray: substrate; shades of yellow: cells). Blue and red circular arcs (of radius $R$) delineate areas of local increase or decrease in the level of regulatory factors, respectively. The following elementary events are depicted: (i) lower cell retracts (two grid sites in row 2); (ii) lower cell protrudes (row 5); (iii) upper cell protrudes (row 10). In addition, the following elementary events occur across the cell-cell boundary: (iv) retraction of upper cell leads to rupture of cell-cell contacts (row 6, right event); (v) either the lower cell protrudes and pushes the upper cell or the upper cell retracts and pulls on the lower cell (row 6, left event). Specifically, event (v) entails mechanical signaling between the upper and lower cell and, therefore, affects the distribution of regulatory factors in both cells. *Right* (**B**): Identical copy of the left image (**A**). Colors indicate local levels of regulatory factors $F$ (blue: $F$ is positive; white: $F$ is zero; red: $F$ is negative; gray: substrate site). Note, in particular, that a substrate grid site has been inserted where cell rupture occurred (row 6, right grid site). The following cases can be distinguished: (i) Grid site $\mathbf{x}_k$ lies in the zone of influence of only positive (blue circles) or negative (red circles) chemical feedback, in which case the level of regulatory factors is positive or negative, respectively (e.g. red grid sites in row 2, or blue grid sites in row 5). (ii) Grid site $\mathbf{x}_k$ lies outside of any zone of influence, in which case the level regulatory factors is zero (e.g. white grid sites in row 2). (iii) Grid site $\mathbf{x}_k$ lies in the zone of influence of equally many positive and negative feedbacks, in which case the level of regulatory factors remains zero (e.g. fourth grid site in row 4). (iv) Grid site $\mathbf{x}_k$ lies in a zone of predominantly positive or negative feedback, in which case the level of regulatory factors is positive or negative, respectively (e.g. third grid site in row 4). Recall that only the sign of $F$ is of significance to update the cells' polarization field; cf. *Equation S13*.

After this update procedure for $\epsilon(\mathbf{x}_n, t)$ is completed, all regulatory factors are reset, $F(\mathbf{x}_n) \to 0, \ \forall n$. This prevents 'spurious memory effects' which may arise once the cell's rear reaches its initial leading edge position as time goes on. In essence, resetting regulatory factors upon completion of one MCS implies that the diffusion-degradation dynamics, underlying the distribution of regulatory factors, is fast on the scale of one MCS.

We emphasize that the polarization field $\epsilon(\mathbf{x}_n)$ is defined only for grid sites $\mathbf{x}_n \in \mathcal{D}^{(\alpha)}$ occupied by an actual cell ($\alpha \geq 0$). To allow for spatial variations of substrate properties, we therefore introduce a second scalar field variable $\varphi(\mathbf{x}_n)$, which is defined on the entire computational grid. The scalar field $\varphi(\mathbf{x}_n)$ is taken to measure the local density of anchoring points that a cell might use to form focal adhesions. Although one might consider to treat $\varphi$ as a time-dependent field variable, in this work $\varphi$ is used to implement static substrate patterns, only. The field $\varphi(\mathbf{x}_n)$ is thus initialized once at the beginning of the simulation and kept fixed throughout the entire simulation.

Having introduced the fields $\epsilon(\mathbf{x}_n)$ and $\varphi(\mathbf{x}_n)$, we now discuss their impact on the system's dynamics by giving their contribution to the goal function $p(\mathcal{T})$. Suppose that the elementary event $\mathcal{T}$ is attempted by a source cell $\alpha$ at source grid site $\mathbf{x}_s \in \mathcal{B}^{(\alpha)}$. Further, let $\mathbf{x}_t$ denote the

target grid site and $\beta$ denote the index of the target cell (where as usual $\beta \geq 0$ indicates that $\mathbf{x}_t$ is occupied by an actual cell, $\beta < 0$ means that $\mathbf{x}_t$ exposes substrate). We then define the *'polarization energy'* $\Delta\mathcal{H}_{\mathrm{cyto}}(\mathcal{T})$ as follows:

$$\Delta\mathcal{H}_{\mathrm{cyto}}(\mathcal{T}) \equiv \begin{cases} \epsilon(\mathbf{x}_t) - \epsilon(\mathbf{x}_s), & \mathcal{T} \,\hat{=}\, \mathcal{T}_{\mathrm{pro}} \,\wedge\, \beta \geq 0, \\ \epsilon(\mathbf{x}_s) - \epsilon(\mathbf{x}_t), & \mathcal{T} \,\hat{=}\, \mathcal{T}_{\mathrm{ret}} \,\wedge\, \beta \geq 0, \\ -[\epsilon(\mathbf{x}_s) + \varphi(\mathbf{x}_t)], & \mathcal{T} \,\hat{=}\, \mathcal{T}_{\mathrm{pro}} \,\wedge\, \beta < 0, \\ \epsilon(\mathbf{x}_s) + \varphi(\mathbf{x}_t), & \mathcal{T} \,\hat{=}\, \mathcal{T}_{\mathrm{ret}} \,\wedge\, \beta < 0, \end{cases} \tag{S14a}$$

Here, the definition of $\Delta\mathcal{H}_{\mathrm{cyto}}$ is such that the likelihood of cell protrusions is enhanced if the concentration of cytoskeletal structures at the source grid site, $\epsilon(\mathbf{x}_s)$, is larger than the concentration at the target grid site, $\epsilon(\mathbf{x}_t)$ (first row of *Equation S14a*), and vice versa for cell retractions (second row of *Equation S14a*). The strength of focal adhesions is taken to be measured by the sum $\epsilon + \varphi$. Their associated 'anchoring effects' (which increase with growing strength of focal adhesions) promote the formation of cell protrusions against unoccupied substrate sites (third row of *Equation S14a*), and, correspondingly, impede cell retractions (fourth row of *Equation S14a*). Note, in particular, that the first two rows of *Equation S14a* can be obtained from a combined evaluation of the lower two rows. For example, if source cell $\alpha$ annexes $\mathbf{x}_t$ starting from $\mathbf{x}_s$, two things need to happen: First, focal adhesions formed by the target cell $\beta$ must be broken, implying a contribution $\Delta\mathcal{H}_{\mathrm{cyto}} = \epsilon(\mathbf{x}_t) + \varphi(\mathbf{x}_t)$ (fourth row of *Equation S14a*). Secondly, new focal adhesions are formed by the source cell $\alpha$, implying a contribution $\Delta\mathcal{H}_{\mathrm{cyto}} = -[\epsilon(\mathbf{x}_s) + \varphi(\mathbf{x}_t)]$ (third row of *Equation S14a*). Taking the sum of both contributions gives the expression in the first row of *Equation S14a*. An analogous line of arguments leads to the expression in the second row of *Equation S14a*.

The contribution to the goal function $p(\mathcal{T})$ due to the *polarization energy* $\Delta\mathcal{H}_{\mathrm{cyto}}(\mathcal{T})$ is then defined by

$$p_{\mathrm{cyto}}(\mathcal{T}) := \exp[-\Delta\mathcal{H}_{\mathrm{cyto}}(\mathcal{T})/k_{\mathrm{B}}T], \tag{S14b}$$

where we set the effective thermal energy to $k_{\mathrm{B}}T \equiv 1$. The characteristic 'energy scale' for $\Delta\mathcal{H}_{\mathrm{cyto}}$ is set by the polarization bounds $\epsilon_0 - \Delta\epsilon/2$ and $\epsilon_0 + \Delta\epsilon/2$, which turns out to have important implications for collective cell dynamics, as discussed in the main text.

## Cell adhesion

To implement the ability of cells to establish cell adhesions across cell-cell interfaces, we use a special form for the respective contribution to the goal function $p$, which is designed to distinguish between the formation of new and the breakage of existing cell-cell adhesion sites.

To this end, we define *adhesion matrices* $B_{\alpha,\beta}$ and $B'_{\alpha,\beta}$ quantifying the system's change in 'energy' upon forming a new contact between cells $\alpha$ and $\beta$ [$B_{\alpha,\beta}$] and upon breaking a pre-existing contact between those cells by an 'intruder cell' $\gamma \neq \alpha, \beta$ [$B'_{\alpha,\beta}$]. In our computational model, we assume that formation of new cell-cell contacts is energetically favored, and that breaking of pre-existing contacts by intruder cells is energetically penalized. The matrix entries of $B_{\alpha,\beta}$ and $B'_{\alpha,\beta}$, therefore, have a definite sign, which we take to be positive. The ordering of the cell index pair of $B_{\alpha,\beta}$ and $B'_{\alpha,\beta}$ is of no physical significance, i.e. the adhesion matrices are symmetric. We also assume that a given cell $\alpha$ does not interact with itself, such that the diagonal elements of the adhesion matrices vanish. Finally, there is no adhesion between cells and empty substrate sites, such that all matrix elements containing a negative cell index vanish. In summary, the adhesion matrices $B_{\alpha,\beta}$ and $B'_{\alpha,\beta}$ exhibit the following properties:

$$B_{\alpha,\beta} = B_{\beta,\alpha} \geq 0, \tag{S15a}$$

$$B'_{\alpha,\beta} = B'_{\beta,\alpha} \geq 0, \tag{S15b}$$

$$B_{\alpha,\alpha} = B'_{\alpha,\alpha} = 0, \tag{S15c}$$

$$B_{\alpha,\beta} = B'_{\alpha,\beta} = 0, \quad \text{if } \alpha < 0 \ \vee \ \beta < 0. \tag{S15d}$$

In addition, we assume that the energy cost associated with the breakage of a given cell-cell contact exceeds the energetic benefit of its initial formation, i.e.

$$B'_{\alpha,\beta} \geq B_{\alpha,\beta}, \tag{S15e}$$

where equality of both quantities would reproduce the assumption underlying the standard cellular Potts model (**Graner and Glazier, 1992**; **Glazier and Graner, 1993**). We shall refer to this property as the 'dissipative nature of cell-cell adhesion'.

To implement the effects of cell-cell adhesion, we compute the 'energy difference' $\Delta\mathcal{H}_{\mathrm{adh}}(\mathcal{T})$ for any given elementary event $\mathcal{T}$ according to the scheme illustrated in **Appendix 1—figure 3**. One has to distinguish between *protrusion* and *retraction* events. First, say that a cell $\alpha$ attempts a protrusion event $\mathcal{T}_{\mathrm{pro}}$, involving the source grid site $\mathbf{x}_s \in \mathcal{B}^{(\alpha)}$ and the target grid site $\mathbf{x}_t \in \mathcal{B}^{(\beta)}$, as illustrated in **Appendix 1—figure 3A**. In this case, cell $\alpha$ acts as intruder cell, since the depicted protrusion event affects three pre-existing contacts between the target cells $\beta$ and a 'third party' cell $\gamma$. Acceptance of the depicted protrusion event would have the following energetically relevant effects: (i) All pre-existing contacts between the target cell $\beta$ and third party cell $\gamma \neq \alpha, \beta$ at the target grid site $\mathbf{x}_t$ are torn apart. (ii) New contacts between the source cell $\alpha$ and third party cell $\gamma \neq \alpha, \beta$ are established. (iii) The length of the cell contact line between source cell $\alpha$ and target cell $\beta$ is changed. Altogether, these three effects lead to the following cell adhesion energy difference,

$$\Delta\mathcal{H}_{\mathrm{adh}}(\mathcal{T}_{\mathrm{pro}}) \equiv \ -\ell \sum_{\mathbf{x}_j \in \mathcal{N}_t} \left[ B_{\alpha,c(\mathbf{x}_j)} - \delta_{\alpha,c(\mathbf{x}_j)} B_{\beta,c(\mathbf{x}_j)} \right]$$
$$+\ell \sum_{\mathbf{x}_j \in \mathcal{N}_t} B'_{\beta,c(\mathbf{x}_j)}(1 - \delta_{\alpha,c(\mathbf{x}_j)}), \tag{S16a}$$

where $\ell$ is the length of a hexagon edge. The first term in this expression accounts for the (energetically favored) formation of new cellular contacts, as well as for the remodeling of the interface between source cell $\alpha$ and target cell $\beta$ [points (ii) and (iii)]. The second term measures the (energetically penalized) breaking of pre-existing cell contacts [point (i)] and, therefore, impedes cell $\alpha$'s ability to intrude. Conversely, if source cell $\alpha$ attempts a retraction event $\mathcal{T}_{\mathrm{ret}}$, then the same reasoning as the one leading to **Equation S16a** applies, only this time the elementary event proceeds in reverse, i.e. from the target site $\mathbf{x}_t$ to the source site $\mathbf{x}_s$; cf. **Appendix 1—figure 3A**:

$$\Delta\mathcal{H}_{\mathrm{adh}}(\mathcal{T}_{\mathrm{ret}}) \equiv \ -\ell \sum_{\mathbf{x}_j \in \mathcal{N}_s} \left[ B_{\beta,c(\mathbf{x}_j)} - \delta_{\beta,c(\mathbf{x}_j)} B_{\alpha,c(\mathbf{x}_j)} \right]$$
$$+\ell \sum_{\mathbf{x}_j \in \mathcal{N}_s} B'_{\alpha,c(\mathbf{x}_j)}(1 - \delta_{\beta,c(\mathbf{x}_j)}), \tag{S16b}$$

where $\ell$ is the length of a hexagon edge.

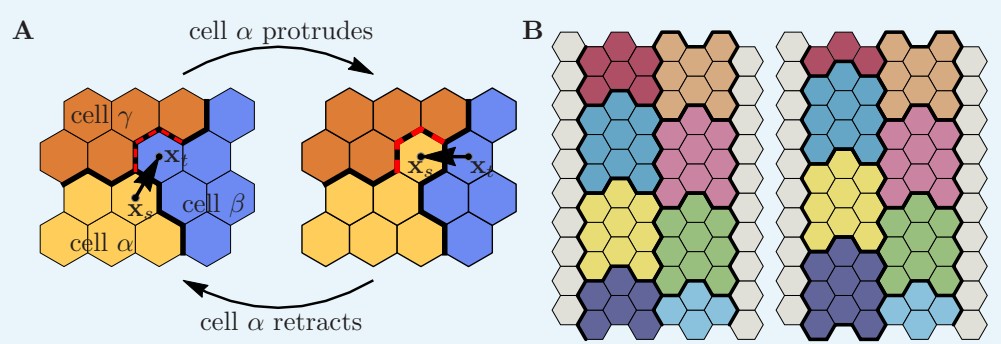

**Appendix 1—figure 3.** Cell-cell adhesion. (**A**) Adhesive energy contribution in a cyclic process, where a protrusion of source cell $\alpha$ against target cell $\beta$ is followed by the inverse retraction event. Both events involve a third party cell $\gamma$, leading to net energy dissipation after the cyclic process has been completed. *Protrusion*: (i) Three pre-existing cell-cell contacts between $\beta$ and $\gamma$ are torn apart (red dashed contacts); (ii) three new contacts between $\alpha$ and $\gamma$ are formed; (iii) the contact length between source cell $\alpha$ and target cell $\beta$ increases by one unit of length. This implies $\Delta\mathcal{H}_{\text{adh}}(\mathcal{T}_{\text{pro}}) = \ell\left(3B'_{\beta,\gamma} - 3B_{\alpha,\gamma} - B_{\alpha,\beta}\right)$. *Retraction*: (i) Three pre-existing cell-cell contacts between $\alpha$ and $\gamma$ are torn apart (red dashed contacts); (ii) three new contacts between $\beta$ and $\gamma$ are formed; (iii) the contact length between source cell $\alpha$ and target cell $\beta$ decreases by one unit of length. This implies $\Delta\mathcal{H}_{\text{adh}}(\mathcal{T}_{\text{ret}}) = \ell\left(3B'_{\alpha,\gamma} - 3B_{\beta,\gamma} + B_{\alpha,\beta}\right)$. Altogether, this leads to $\Delta\mathcal{H}_{\text{adh}}^{(\text{cycl})} = \Delta\mathcal{H}_{\text{adh}}(\mathcal{T}_{\text{pro}}) + \Delta\mathcal{H}_{\text{adh}}(\mathcal{T}_{\text{ret}}) = \ell\left(3(\Delta B)_{\alpha,\gamma} + 3(\Delta B)_{\beta,\gamma}\right) \geq 0$, i.e. a (non-negative) dissipative contribution, whose magnitude depends on the dissipation matrix $(\Delta B)_{\alpha,\beta} = B'_{\alpha,\beta} - B_{\alpha,\beta} \geq 0$. (**B**) Shear viscosity due to cell-cell adhesion. Consider two rows of adhesive cells sliding past each other as indicated in the figure (left row of cells moves up by one grid site; colors indicate different cells). The associated adhesion energy change (per cell) reads $\Delta\mathcal{H}_{\text{adh}}/n_c = 2\left(B' - B\right) \geq 0$, where $n_c$ denotes the number of cells sliding past each other, and where we assumed cells of like type, i.e. $B_{\alpha,\beta} \equiv B$ and $B'_{\alpha,\beta} \equiv B'$ ($\alpha \neq \beta$). The condition $B'>B$, **Equation S15e**, thus implies positive friction associated with cellular shear flows, whose magnitude is proportional to the number of cells sliding past each other. Note that this shear viscosity vanishes for $B' = B$, i.e. for zero dissipation matrix.

We may now use **Equation S16a** and **Equation S16b** to illustrate the 'dissipative nature' of adhesive interactions by means of two archetypical examples. First, consider the adhesive energy contribution to any cyclic process. By a cyclic process we mean a sequence of two mutually inverse elementary events, e.g. a protrusion event $\mathcal{T}_{\text{pro}}$, which is immediately followed by its inverse retraction event $\mathcal{T}_{\text{ret}}$, such that the system's final configuration is identical to its initial configuration. Using **Equation S16** we find for the total adhesive energy contribution to a cyclic process:

$$\Delta\mathcal{H}_{\text{adh}}^{(\text{cycl})} = \ell \sum_{\mathbf{x}_j \in \mathcal{N}_t \backslash (\mathcal{D}^{(\alpha)} \cup \mathcal{D}^{(\beta)})} \left[(\Delta B)_{\alpha,c(\mathbf{x}_j)} + (\Delta B)_{\beta,c(\mathbf{x}_j)}\right], \tag{S17}$$

$$(\Delta B)_{\alpha,\beta} := B'_{\alpha,\beta} - B_{\alpha,\beta} \geq 0, \tag{S18}$$

and can therefore conclude that

$$\Delta\mathcal{H}_{\text{adh}}^{(\text{cycl})} \geq 0,$$

where $\mathcal{N}_t$ denotes the neighborhood of the grid site which temporarily changes its cell index, and where $\alpha$ and $\beta$ are the indices of the source and target cells involved in the cyclic process; cf. **Appendix 1—figure 3A**. Since $(\Delta B)_{\alpha,\beta} \geq 0$, the above adhesive energy contribution is non-negative, thus leading to an amount of energy equal to $\Delta\mathcal{H}_{\text{adh}}^{(\text{cycl})}$ being dissipated as the cyclic process completes. This leads us to refer to the parameter matrix $(\Delta B)_{\alpha,\beta}$ as *dissipation matrix*. Second, consider two (infinitely extended) columns of cells in

adhesive contact, sliding past each other. This situation is depicted in *Appendix 1—figure 3B*, where the left column of cells moves (as a whole) upwards by one grid site, while the right column of cells remains stationary. To assess the adhesive energy contribution along the contact line connecting both cell columns, note that the depicted transformation can be implemented by letting each cell in the left column protrude its leading (i.e. upper) edge by one grid site. For each protruding (source) cell $\alpha$, this transformation entails to the following energetic effects (cf. discussion above): (i) Two of the pre-existing cell-cell contacts between the source cell's upper neighbor in the left column (target cell $\beta$) and the corresponding cell in the right column (third party cell $\gamma$) get torn apart, leading to an energetic contribution $2B'_{\beta,\gamma}$. (ii) In return, two new contacts between the protruding (source) cell $\alpha$ and cell $\gamma$ are being established, leading to a contribution $-2B_{\alpha,\gamma}$. (iii) Since the length of the contact line between cells in the left column (i.e. between protruding source cell $\alpha$ and retracting target cell $\beta$) remains unchanged, there's no further energetic contribution due to adhesive contacts between cells in the left column. Assuming that all cells in the system are of equal types, we write $B_{\alpha,\beta} \equiv B$ and $B'_{\alpha,\beta} \equiv B'$ ($\alpha \neq \beta$), and therefore, find

$$\Delta\mathcal{H}^{(\text{visc})}_{\text{adh}} = 2\ell\,(B' - B) \equiv 2\ell\,\Delta B \geq 0, \tag{S19}$$

i.e. a non-negative dissipative contribution per cell. The size of the dissipation parameter $\Delta B$ thus introduces a natural means to tune the system's *shear viscosity*.

With the above definitions of the adhesive energy changes, *Equation S16*, we define the contribution of cell adhesion to the goal function $p(\mathcal{T})$ as follows:

$$p_{\text{adh}}(\mathcal{T}) := \exp\left[-\Delta\mathcal{H}_{\text{adh}}(\mathcal{T})/k_{\text{B}}T\right], \tag{S20}$$

where we set the effective thermal energy to $k_{\text{B}}T \equiv 1$.

## Rupture of cell contacts

By now, we have introduced all components making up the total acceptance probability $p(\mathcal{T})$, *Equation S9*. To conclude our discussions concerning the implementation of cellular traits, we highlight one additional aspect of elementary events. So far, the notion of an elementary event can be summarized as follows: Once source and target grid sites, $\mathbf{x}_s$ and $\mathbf{x}_t$, have been selected, acceptance of a protrusion [retraction] event causes (among other things like the distribution of regulatory factors) the cell index to be copied from $\mathbf{x}_s$ to $\mathbf{x}_t$ [from $\mathbf{x}_t$ to $\mathbf{x}_s$]. In other words, the domain $\mathcal{D}^{(\alpha)}$ of source cell $\alpha$ annexes $\mathbf{x}_t$ [loses $\mathbf{x}_s$], while the domain $\mathcal{D}^{(\beta)}$ of target cell $\beta$ is forced to let go $\mathbf{x}_t$ [accommodate $\mathbf{x}_s$]. However, if both source and target cells are actual cells, i.e. $\alpha, \beta \geq 0$, and if the source cell attempts a retraction event, there is one additional possible outcome: If cell cohesion is weak, then the pulling force exerted by the retracting source cell $\alpha$ on its neighboring cells might also result in rupture of all pre-existing contacts between the retracting source cell and its neighboring cells at $\mathbf{x}_s$, rather than forcing one of its neighboring cells (the target cell) to fill the void created at $\mathbf{x}_s$ once $\alpha$ retracts; cf. rupture event depicted in *Appendix 1—figure 2*. To test for the occurrence of cell rupture, the total energetic cost of each attempted retraction event between two actual cells is evaluated under two different assumptions: First, we assume that the pulling force exerted by the retracting source cell $\alpha$ on target cell $\beta$ is strong enough to force $\beta$ to fill the void created at $\mathbf{x}_s$ (i.e. to accommodate $\mathbf{x}_s$), and call this a regular retraction event $\mathcal{T}_{\text{ret}}$. Secondly, we assume that the retraction of source cell $\alpha$ causes all pre-existing cell-cell contacts of cell $\alpha$ at $\mathbf{x}_s$ to rupture, leaving a free substrate site at $\mathbf{x}_s$ after the retraction event has been accepted. This latter event will be referred to as rupture event $\mathcal{T}_{\text{rup}}$. We then compute the total energy differences

$$\Delta\mathcal{H}(\mathcal{T}_{\text{ret}}) = \Delta\mathcal{H}_{\text{cont}}(\mathcal{T}_{\text{ret}}) + \Delta\mathcal{H}_{\text{cyto}}(\mathcal{T}_{\text{ret}}) + \Delta\mathcal{H}_{\text{adh}}(\mathcal{T}_{\text{ret}})$$

and

$$\Delta \mathcal{H}(\mathcal{T}_{\mathrm{rup}}) = \Delta \mathcal{H}_{\mathrm{cont}}(\mathcal{T}_{\mathrm{rup}}) + \Delta \mathcal{H}_{\mathrm{cyto}}(\mathcal{T}_{\mathrm{rup}}) + \Delta \mathcal{H}_{\mathrm{adh}}(\mathcal{T}_{\mathrm{rup}})$$

under both assumptions (the energy difference associated with accepting $\mathcal{T}_{\mathrm{rup}}$ can be computed with **Equation S16b** by using the substrate $\beta = -1$ as new target cell) and compare the respective outcomes. If the rupture event is energetically favored over the regular retraction event, i.e. $\Delta \mathcal{H}(\mathcal{T}_{\mathrm{rup}}) < \Delta \mathcal{H}(\mathcal{T}_{\mathrm{ret}})$, then cohesion between cells is weak. In this case, the rupture event $\mathcal{T}_{\mathrm{rup}}$, rather than the regular retraction event $\mathcal{T}_{\mathrm{ret}}$ will be attempted. Otherwise, cohesion is strong and a regular retraction event $\mathcal{T}_{\mathrm{ret}}$ will be attempted.

### Rupture of substrate contacts

In our discussion so far, a cyclic process that follows up a protrusion event $\mathcal{T}_{\mathrm{pro}}$ with its inverse retraction event $\mathcal{T}_{\mathrm{ret}}$, involving a cell $\alpha$ and no third party cells (in other words: no cell-cell contacts are made or broken), will not yield a net energy cost or gain; cf. **Equation S14a**. To account for the dissipative nature of cell-substrate contacts, we proceed similarly as when we have considered the dissipative nature of cell-cell contacts. We introduce dissipation in substrate adhesion by leaving the Hamiltonian unaltered for protrusion events but adding a penalty for retraction events:

$$\Delta \mathcal{H}(\mathcal{T}_{\mathrm{ret}}) \rightarrow \Delta \mathcal{H}(\mathcal{T}_{\mathrm{ret}}) + D. \tag{S21}$$

Therefore, a cell that adheres to the substrate at some grid point has to pay a cost $D$ to retract from it. In other words, we assume that a fixed amount of energy $D$ is dissipated once the adhesive bonds between a cell and the substrate break.

To keep its overall size across translations, the cell has to gain and lose equal amounts of hexagons, with $\Delta \epsilon$ as the maximal energy gained by a single gain-and-loss in the absence of dissipation. In the presence of dissipation however, the cell has to pay at least a cost of $(\epsilon_0 - \Delta \epsilon / 2) + D$ to detach at an arbitrary location, resulting in $\Delta \epsilon - D$ as the maximal energy gained by a single gain-and-loss in the presence of dissipation. Thus, while for $D = 0$ there is no impact of substrate dissipation on cell motility, it will at the latest for $D = \Delta \epsilon$ completely inhibit (directed) cell migration. Therefore, we study substrate dissipation in the range $D \in [0, \Delta \epsilon]$.

### Cell domain update routine

Having discussed the implementation concerning the two basic types of elementary events, namely protrusion events $\mathcal{T}_{\mathrm{pro}}$ and retraction events $\mathcal{T}_{\mathrm{ret}}$, as well as the two subtypes of regular retraction events and rupture events $\mathcal{T}_{\mathrm{rup}}$, we can now summarize the cell domain update routine, point 3.5 in section 'Monte-Carlo scheme'. To this end, and in accordance with our previous notation, we use the cell indices $\alpha$ and $\beta$ to denote source and target cell, and $\mathbf{x}_s$ and $\mathbf{x}_t$ to denote source and target grid site. Moreover, equal signs "=" in the following listing are to be interpreted as assignment operators, where the value of the variable on the right hand side of the operator is assigned to the variable on the left hand side. With these preliminary remarks in mind, the cell update routine can be summarized as follows:

- If the accepted elementary event is a protrusion event:
  1. Set $\epsilon(\mathbf{x}_t) = \epsilon(\mathbf{x}_s)$ and $F(\mathbf{x}_t) = F(\mathbf{x}_s)$.
  2. $\mathcal{D}^{(\alpha)} \rightarrow \mathcal{D}^{(\alpha)} \cup \{\mathbf{x}_t\}$.
  3. $\mathcal{D}^{(\beta)} \rightarrow \mathcal{D}^{(\beta)} \setminus \{\mathbf{x}_t\}$.
  4. Distribute regulatory factors according to **Equation S12a**.
- If the accepted elementary event is a regular retraction event:
  1. Set $\epsilon(\mathbf{x}_s) = \epsilon(\mathbf{x}_t)$ and $F(\mathbf{x}_s) = F(\mathbf{x}_t)$.
  2. Set $\mathcal{D}^{(\alpha)} \rightarrow \mathcal{D}^{(\alpha)} \setminus \{\mathbf{x}_s\}$
  3. Set $\mathcal{D}^{(\beta)} \rightarrow \mathcal{D}^{(\beta)} \cup \{\mathbf{x}_s\}$
  4. Distribute regulatory factors according to **Equation S12b**.
- If the accepted elementary event is a rupture event:
  1. Set $\epsilon(\mathbf{x}_s) = 0$ and $F(\mathbf{x}_s) = 0$.
  2. Set $\mathcal{D}^{(\alpha)} \rightarrow \mathcal{D}^{(\alpha)} \setminus \{\mathbf{x}_s\}$

3.  Distribute regulatory factors according to *Equation S12c*.

## Cell proliferation and mitosis

While cell proliferation and mitosis play no role in the experimental setup of rotating cell clusters, cell growth and division are observed experimentally in a setup where a sheet of cells expands into free space after removal of a stencil. Therefore, it is essential to include proliferation of cells in the numerical model. How this is done is described in this section.

We distinguish between two phases in the cell cycle, an *interphase* during which cells roughly double in volume and *mitosis*, the process of cell division. Even though a further partitioning of the interphase was considered in previous work (*Li and Lowengrub, 2014*), we do not expect that such a distinction is relevant for our results. In our computational framework cell growth may be implemented by progressively changing any cellular parameter that affects the cell's equilibrium size. The two possible, largely equivalent choices are a successive reduction of the area coupling constant $\kappa_A^{(\alpha)}$ or an increase of the average cell polarization $\epsilon_0^{(\alpha)}$. We here employ the first method. We assume that individual cells grow exponentially (*Barber et al., 2017*) over a well-defined period $T_g$. Additional variability in cell cycle length can be achieved by introducing an additional refractory phase with exponentially distributed waiting times and the average waiting time $T_r$, which we set to $T_r = 0$ in this work. Moreover, we assume that the migratory behavior of a cell should not change significantly as it grows. However, as the cell grows in size by a factor of $2$, it also increases its perimeter and the corresponding energy cost for adding new membrane segments roughly by a factor of $\sqrt{2}$. Therefore, as we do not scale the polarization field $\epsilon$ and the resulting energy gains for protrusions during cell growth, we mitigate the increased cost for ruffling the membrane by reducing the perimeter stiffness by a factor of $\sqrt{2}$. The quantitative viability of this approach is further discussed in section 'Single cell size'.

To prevent tissue overgrowth, cell proliferation is generally contact inhibited in healthy cells: When the tissue approaches a state where each cell has formed adhesive contacts with the substrate and is completely surrounded by neighbours, cells stop proliferating. In addition, it has been proposed that the pressure or local density in the tissue has a negative impact on the local growth rate (*Shraiman, 2005*; *Ranft et al., 2010*). To account for these phenomena in the model, we complement the two cell cycle periods interphase and mitosis by a quiescent cell state during which cell growth is halted. The parameters $\kappa_A$ and $\kappa_P$ are, therefore, kept constant for a quiescent cell; we denote the corresponding values as $\kappa_{A/0}$ and $\kappa_{P/0}$. There are many possible ways to implement contact inhibition in our computational model. For example, it could be implemented by allowing a quiescent cell to enter the cell cycle triggered by low local cell density, or when a sufficiently large fraction of its membrane length is not in contact with neighbour cells but exposed to free space. In our model it proves numerically advantageous to make a quiescent cell enter the interphase when its area succeeds a certain reference area. We choose this area threshold as $A_T = r A_{\text{ref}}$, where the factor $r = \mathcal{O}(1)$ relates the threshold size to the equilibrium cell size $A_{\text{ref}}$ reached by a free, solitary cell with constant polarization field $\epsilon = \epsilon_0$. Cells living in a densely packed environment will not exceed the area threshold due to the pressure exerted on them by neighboring cells and can, therefore, not grow. Conversely, cells exposed to free space are more likely to reach this threshold and proliferate. Finally, a growing cell in interphase becomes mitotic after the growth time $T_g$ has passed, at which point cell size has roughly doubled with respect to the size in the quiescent period. We assume that cell migration and mitosis are processes that exclude each other. Hence, the positive feedback leading to persistent cell migration is switched off for mitotic cells and the polarization field relaxes to the neutral state $\epsilon_0$ according to *Equation S13c*.

There appears to be no universal set of rules which determine the orientation of the cleavage plane along which cells divide (*Minc and Piel, 2012*). Rather, for epithelial tissues there are a variety of factors which include local cell geometry and the direction of stress in the tissue (*Gibson and Gibson, 2009*). Though it is in principle possible to implement any given rule in our computational model, in its present version the axis along which a cell divides

is chosen as a random direction through the geometric center of the cell. In case of irregular cell shapes, a separation of the cellular domain into more than two connected components can occur. To prevent violation of topological constraints, in this case the two largest components are considered as descendant cells and the residual grid sites are filled by substrate.

We explicitly account for the finite duration of the mitotic phase $T_d$ by keeping the cells in a mitotic state for the aforementioned time period, until the final instantaneous splitting of the cellular domains. After cell division, persistent cell migration of the daughter cells is enabled again. The descendent cells will subsequently re-enter the growing phase if their area exceeds the defined threshold, as mentioned above.

The following list summarizes the steps motivated and explained in the previous paragraphs. These additional steps are performed in a simulation that includes cell proliferation:

1. Assign a state variable $s^{(\alpha)}$ to each cell which encodes the current phase in the cell cycle:

$$s^{(\alpha)} = \begin{cases} 0, & \text{quiescent phase} \\ 1, & \text{refractory phase} \\ 2, & \text{interphase} \\ 3, & \text{mitotic phase} \end{cases} \tag{S22}$$

2. Compute the equilibrium size $A_{\mathrm{ref}} = (\epsilon_0 - 2\pi\sqrt{3}\kappa_P)/(\sqrt{3}\kappa_A)$ of a free, solitary cell with fixed polarization field $\epsilon = \epsilon_0$, which spreads on the substrate used in the simulation.

3. At the beginning of the simulation, $t = 0$, all cells are in the quiescent state, $s^{(\alpha)}(0) = 0$, and have the following area and perimeter coupling constants, respectively: $\kappa_A^{(\alpha)}(0) = \kappa_{A/0}$ and $\kappa_P^{(\alpha)}(0) = \kappa_{P/0}$.

4. After the completion of each Monte Carlo time step $t$, perform one of the following changes for each cell:

• Switch from quiescent to refractory state:

$$s^{(\alpha)}(t) = 0 \,\wedge\, A^{(\alpha)}(t) > A_{\mathrm{T}}$$
$$\Rightarrow s^{(\alpha)}(t+1) = 1. \tag{S23}$$

• Switch from refractory state to growing state with probability $p = 1 - \exp(-1/T_r)$:

$$s^{(\alpha)}(t) = 1$$
$$\Rightarrow s^{(\alpha)}(t+1) = \begin{cases} 2, & (p), \\ 1, & (1\text{-}p), \end{cases} \tag{S24}$$

where the terms in the brackets denote the respective probability.

• Exponential growth in interphase over a period of $T_g$:

$$s^{(\alpha)}(t) = 2$$
$$\Rightarrow \kappa_A^{(\alpha)}(t+1) = \kappa_A^{(\alpha)}(t) \cdot (1/2)^{1/T_g}$$
$$\Rightarrow \kappa_P^{(\alpha)}(t+1) = \kappa_P^{(\alpha)}(t) \cdot (1/2)^{1/(2T_g)}. \tag{S25}$$

• Switch from interphase to mitosis:

$$s^{(\alpha)}(\tau) = 2 \text{ for all } \tau \in [t - T_g, t]$$
$$\Rightarrow s^{(\alpha)}(t+1) = 3. \tag{S26}$$

- During cell division, cell motility is switched off and the polarization field relaxes to the neutral state according to *Equation S13c*.
- Perform cell division, reset area and perimeter stiffness and exit mitotic phase:

$$
\begin{aligned}
&s^{(\alpha)}(\tau) = 3 \text{ for all } \tau \in [t - T_d,\, t] \\
&\Rightarrow \text{ divide cell } \alpha \text{ into cells } (\alpha, \beta) \\
&\Rightarrow \kappa_A^{(\alpha,\beta)}(t+1) = \kappa_{A/0} \\
&\Rightarrow \kappa_P^{(\alpha,\beta)}(t+1) = \kappa_{P/0} \\
&\Rightarrow s^{(\alpha,\beta)}(t+1) = 0.
\end{aligned} \tag{S27}
$$

- Cell motility is restored after cell division.
- If none of the above rules apply, then do not perform any changes.

## Numerical computation of stress in a tissue

In the section describing the numerical results on tissue expansion, the stress distribution in the tissue is shown in the kymographs *Figure 5(C,G)* and *Figure 6(C,G)*. Hereafter we explain how the stress tensor for each cell in the tissue can be computed from the forces acting on the cell's membrane segments in the Monte Carlo simulation. The mean value of the stress tensor in a deformed body can be calculated numerically from the formula

$$
\bar{\sigma}_{ij}^{(\alpha)} = \frac{\ell}{2A^{(\alpha)}} \sum_{\mathbf{x}_k \in \mathcal{B}^{(\alpha)}} \left( f_k^i \tilde{x}_k^j + f_k^j \tilde{x}_k^i \right), \tag{S28}
$$

which is a discretized version of the surface integral in *Landau et al. (1986)*. Here, $\mathbf{f}_k$ is the force acting on the membrane element $\mathbf{x}_k$ of cell $\alpha$, $\tilde{\mathbf{x}}_k = \mathbf{x}_k - \mathbf{x}_{\mathrm{com}}^{(\alpha)}$ is the position of the element with respect to the center of mass $\mathbf{x}_{\mathrm{com}}^{(\alpha)}$ of the cell, and the superscripts $i$ and $j$ are Cartesian indices. The forces $\mathbf{f}_k$ can be computed from the energy differences of all possible protrusion and retraction events originating from $\mathbf{x}_k$,

$$
\begin{aligned}
\mathbf{f}_k = \; &- \sum_{\mathbf{x}_l \in \mathcal{N}_k} \frac{\Delta \mathcal{H}(\mathcal{T}_{\mathrm{pro}})}{\|\mathbf{x}_l - \mathbf{x}_k\|} \frac{\mathbf{x}_l - \mathbf{x}_k}{\|\mathbf{x}_l - \mathbf{x}_k\|} \\
&- \sum_{\mathbf{x}_l \in \mathcal{N}_k}^{\#} \frac{\Delta \mathcal{H}(\mathcal{T}_{\mathrm{ret}})}{\|\mathbf{x}_k - \mathbf{x}_l\|} \frac{\mathbf{x}_k - \mathbf{x}_l}{\|\mathbf{x}_k - \mathbf{x}_l\|},
\end{aligned} \tag{S29}
$$

where the number sign indicates a sum over substrate grid sites only, i.e. grid sites with $c(\mathbf{x}_l) < 0$, and where $\Delta \mathcal{H} \equiv \mathcal{H}_{\mathrm{cont}} + \mathcal{H}_{\mathrm{adh}} + \mathcal{H}_{\mathrm{cyto}}$.

## Numerical computation of the cell shape

We use two complementary measures for the cell shape. The first is a simple measure for the deviation of an object from a circle (we refer to this as *cell extension*):

$$
K = 1 - \frac{4\pi A}{P^2}. \tag{S30}
$$

It becomes zero if the object is a circle and becomes 1 if the object is a line. The second measure for the cell shape is obtained from a principle components analysis of the cell shape. Specifically, we compute the covariance matrix of the point cloud representing the cell domain $\mathcal{D}^{(\alpha)}$:

$$
\left( \mathrm{Cov}(\mathcal{D}^{(\alpha)}) \right)_{ij} = \frac{\sum_{\mathbf{x}_k \in \mathcal{D}^{(\alpha)}} \tilde{x}_k^i \tilde{x}_k^j}{\sum_{\mathbf{x}_k \in \mathcal{D}^{(\alpha)}} 1}, \tag{S31}
$$

where $\tilde{\mathbf{x}}_k = \mathbf{x}_k - \mathbf{x}_{\mathrm{com}}^{(\alpha)}$ denotes the coordinates of element $\mathbf{x}_k$ of cell $\alpha$, relative to the cell's

center of mass $\mathbf{x}_{\text{com}}^{(\alpha)}$; the superscipts $i$ and $j$ are Cartesian indices. Then, we compute the two eigenvalues $l_+^2$ (larger eigenvalue) and $l_-^2$ (smaller eigenvalue) of the covariance matrix, which determine the variance of the point distributions along the two principal axes of the cell. Finally, the aspect ratio of the cell is given by $l_+/l_-$.

## Appendix 2

### Parameter screening in silico

In this section we provide additional analysis of the model parameters beyond what is already shown in the main text.

We explore all three rotational phases $\mathcal{R}_1$, $\mathcal{R}_2$ and $\mathcal{R}_3$ within confinements of varying size and constant cell density. In the $\mathcal{R}_1$-phase, the cell clusters rotate slowly and frequently reorient their direction of rotation. With increasing cell count, the cell clusters cease to rotate. In the highly coordinated $\mathcal{R}_2$ and $\mathcal{R}_3$-phases, we find scale-free behavior such that there is always a macroscopic rotation of the whole cell population regardless of the cell count and corresponding confinement size.

We also explore the parameter space of the tissue simulations. There, we find that an increased cell-cell dissipation $\Delta B$ impairs monolayer growth, while at the same time increasing front roughness. Similarly, an increased cell-substrate dissipation $D$ also impairs monolayer growth. In contrast, increasing the maximum cell polarity $\Delta \epsilon$ improves monolayer growth and also increases front roughness. We thus find that the speed of monolayer expansion depends on whether it is dominated by cell migration or cell proliferation, with the former improving monolayer growth through a better exploration of the cell-free area.

### Single cell size

To rationalize our choice of the cell growth algorithm (see section 'Cell proliferation and mitosis'), we have explored the shape and motility of differently sized solitary cells. To this end, we have varied the area stiffness parameter $\kappa_A$ for different values of perimeter stiffness $\kappa_P$, while keeping all other parameters constant. We find that the area occupied by the motile cell increases linearly with $1/\kappa_A$ (*Appendix 2—figure 1A*). In particular, the cell area can be approximated quite well by the area of an immotile and equilibrated cell with uniform $\epsilon = \epsilon_0$ (fit not shown):

$$A = \frac{\epsilon_0 - 2\pi\sqrt{3}\kappa_P}{\sqrt{3}\kappa_A} \propto 1/\kappa_A. \tag{S32}$$

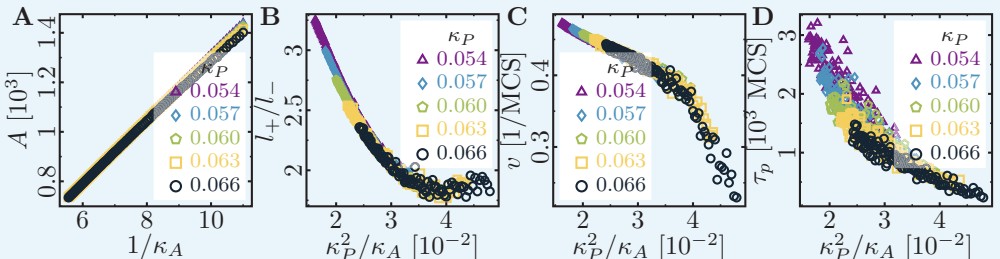

**Appendix 2—figure 1.** Role of area stiffness $\kappa_A$ for cell size and motility. (**A**) The cell area increases linearly with $1/\kappa_A$. The aspect ratio (**B**), speed (**C**) and persistence (**D**) of the cell decrease with increasing cell size. In the simulations, the area elasticity was varied in the interval $\kappa_A \in [0.09, 0.18]$, and the membrane elasticity was chosen from $\kappa_P \in \{0.054, 0.057, 0.060, 0.063, 0.066\}$. Fixed parameters: average cell polarization field $\epsilon_0 = 225$; maximum cell polarity $\Delta\epsilon = 50$; signaling radius $R = 5$; cytoskeletal update rate $\mu = 0.1$; cell-substrate dissipation $D = 0$; cell-substrate adhesion penalty $\varphi = 0$.

Furthermore, we find that with increasing size, and all other parameters constant, cells become rounder, slower, and less persistent (*Appendix 2—figure 1B-D*). To intuitively explain this phenomenology, let us compare a cell of size $A_{\text{ref}}$ with a cell of size $r A_{\text{ref}}$, where $r \in [1, 2]$, with the respective area stiffnesses $\kappa_A$ and $\kappa_A/r$. While the smaller cell has a perimeter $P_{\text{ref}}$, neglecting geometric effects the larger cell has a larger perimeter of approximately $\sqrt{r}P_{\text{ref}}$. Hence, the larger cell has to pay a larger energy cost (roughly by a factor $\sqrt{r}$) to ruffle its

membrane by adding segments and therefore increasing its perimeter. Meanwhile, both the energy gain from the polarization field $\epsilon$ and the energy cost for increasing the cell area (as $A\,\kappa_A = \mathrm{cst.}$) are the same for both cells. Due to the increased energy cost for adding membrane segments, larger cells finds find it more difficult to polarize, and are therefore rounder, slower and less persistent.

To offset this increased energy cost for adding membrane segments to the cell, one can scale the perimeter stiffness of the larger cell to $\kappa_P/\sqrt{r}$, such that $P\,\kappa_P \approx \mathrm{cst}$. We would then predict that the ratio $\kappa_P^2/\kappa_A$ is constant for differently sized cells of similar shape, speed and persistence time. The same relation can also be obtained by realizing that different amounts of grid sites occupied by two otherwise identical cells (in terms of their corresponding Hamiltonian) simply stem from a different discretization of said cells, which is controlled by the parameter $\kappa_A$. Interestingly, we observe such a data collapse for the aspect ratio $l_+/l_-$ and the velocity $v$ of the cells onto two respective master curves depending on the ratio $\kappa_P^2/\kappa_A$ (**Appendix 2—figure 1B,C**). While the proposed data collapse for the persistence time of directed migration of a cell (**Appendix 2—figure 1D**) is somewhat unsatisfactory, this may be owed to the following effect: by keeping $R$ constant we have actually varied the ratio between the area that the signaling molecules typically explore due to diffusion and the area of the cell, $R^2/A$. Finally, we speculate that all observed quantities collapse unto respective master curves $f(\Delta\epsilon\sqrt{\kappa_A}/\kappa_P) \cdot g(R\sqrt{\kappa_A})$.

## Single cell shape and dynamics depend on substrate dissipation

We have also studied the effect of cell-substrate dissipation (see section 'Rupture of substrate contacts') on cell morphology and motility. We have varied the substrate dissipation $D$ for different values of maximum cell polarity $\Delta\epsilon$ and cell perimeter stiffness $\kappa_P$; however, we were not able to achieve a data collapse in $D$ (**Figure 2—figure supplement 1**). We observe that with increasing cell-substrate dissipation, cells become round and cease migrating. This can be illustrated as follows: Consider a situation where the cell conquers a new hexagon at its prospective leading edge. Because the cell on average tends to constrain its area and perimeter while migrating, it consequently needs to lose a different hexagon at its prospective trailing edge. However, this retraction at the trailing edge is energetically penalized and thus cell displacement from its initial position and the positive feedback leading to cell polarization are effectively inhibited. With increasing cell-substrate dissipation, retraction events are further penalized and the cell 'sticks' to the substrate at its trailing edge, preventing persistent motion of the cell. Additionally, to further illustrate the correlation between cell shape and cell migration, we have replotted the values of **Figure 2—figure supplement 1A-C and E-G** (**Figure 2—figure supplement 1D and H**, respectively). Here and in the main text, we find that only cells with an aspect ratio larger than 2 are motile (**Figure 2F**, **Figure 2—figure supplement 1D, H**; white regions).

## Cells in circular confinement

In this section we report on additional parameter studies of the dynamics of cells enclosed in a circular confinement (**Figure 4—figure supplement 1**, **Figure 4—figure supplement 2** and **Figure 4—figure supplement 3**). Specifically, we investigate how the radius of circular confinement affects the synchronized rotation of the cell population. We performed simulations with a densely populated circular confinement and varied the confinement radius, while keeping the cell density constant. The parameters are chosen such that a population consisting of 4 cells (cf. main text, **Figure 4**) would rotate in the $\mathcal{R}_1$, $\mathcal{R}_2$ or $\mathcal{R}_3$-phase, respectively. We studied the mean angular velocity

$$\omega(t) = \hat{\mathbf{e}}_z \cdot \left\langle \frac{\hat{\mathbf{v}}_\alpha(t) \times \tilde{\mathbf{R}}_\alpha(t)}{\|\tilde{\mathbf{R}}_\alpha(t)\|^2} \right\rangle_C , \tag{S33}$$

averaged over the set $\mathcal{C} = \{\alpha \mid \text{is not substrate}\}$ of all cells in confinement, where $\tilde{\mathbf{v}}_\alpha(t) = \mathbf{v}_\alpha(t) - \langle \mathbf{v}_\alpha(t) \rangle_\mathcal{C}$ and $\tilde{\mathbf{R}}_\alpha(t) = \mathbf{R}_\alpha(t) - \langle \mathbf{R}_\alpha(t) \rangle_\mathcal{C}$ are the velocity and position of cell $\alpha$ relative to the cell cluster, respectively.

In the lowly polarizable $\mathcal{R}_1$-phase, small cell populations rotate in a highly synchronized way, and rotation is maximized for populations of 7 cells per confinement (*Figure 4—figure supplement 1A*). As can be inferred from the time traces and snapshots (*Figure 4—figure supplement 1B, C*), cells in small populations all synchronously move in the same direction at a given time and randomly switch between clockwise and counter-clockwise rotation; the switching rate decreases with increasing size of the cell population. Upon increasing the cell count and concomitantly the confinement size, global rotation of the cell population gradually vanishes (*Figure 4—figure supplement 1A*).

Unlike in the $\mathcal{R}_1$-phase, we observe that in the highly polarizable $\mathcal{R}_2$ and $\mathcal{R}_3$-phases populations of all sizes rotate in a highly synchronized way (*Figure 4—figure supplement 2A* and *Figure 4—figure supplement 3A*). There, the dependence of $\langle |\omega| \rangle$ on the population size $N$ can be fitted by a power law of the form $\langle |\omega| \rangle \propto N^{-1/2} \propto r_0^{-1}$. This inverse proportionality between the average angular velocity and the confinement size $r_0$ implies total rotational order, with every cell moving at a constant velocity $|\mathbf{v}_{\text{rot}}| \approx 0.008$ ($\mathbf{v}_{\text{rot}} \perp \tilde{\mathbf{R}}$). Furthermore, in the $\mathcal{R}_2$, and $\mathcal{R}_3$-phases we have only scarcely observed switching of the rotational direction of cell clusters; e.g. for 4-cell clusters in the $\mathcal{R}_3$-phase.

Interestingly, fluctuations in the angular velocity ($\sigma_\omega$) change in a highly non-monotonic fashion with the cell count and concomitantly the confinement size. Certain cell counts exhibit especially high fluctuations of the mean angular velocity (e.g. 5 cells in the $\mathcal{R}_1$-phase, see *Figure 4—figure supplement 1A*; 3 or 10 cells in the $\mathcal{R}_2$-phase, see *Figure 4—figure supplement 2A*; 3 cells in the $\mathcal{R}_3$-phase, see *Figure 4—figure supplement 3A*). This can likely be attributed to frustration of the cells in the population center (*Segerer et al., 2015*); cf. 10 cells in *Figure 4—figure supplement 2C*.

## Velocity and roughness of spreading tissue

We have studied the velocity and roughness of spreading tissue, while varying cell-cell dissipation $\Delta B$, cell-substrate dissipation $D$ and maximum cell polarity $\Delta \epsilon$.

First, let us introduce the observables that we are interested in. Let $X_{>/<}$ be the sets of $x$-coordinates of the left and right outermost edges of the cell sheet. Our *in silico* setup is axially symmetric with respect to the $y$-axis. This initial symmetry persists, as the cell fronts advance towards the cell-free area with the same average speed. Hence, it is not needed to consider the two cell fronts separately, and we can instead consider the set of *unsigned* front positions $X := \text{abs}(X_{>/<})$. Then, we define the average front position as $x_\text{F} := \mathbb{E}(X)$ and the front roughness as $\sigma_\text{F}^2 := \text{Var}(X)$. In particular, we study the total growth of the tissue over the course of 500 MCS, which is captured by the maximal position of the front, $\max(x_\text{F})$, as well as the maximal roughness of the front $\max(\sigma_\text{F})$. We have chosen our parameters such that a cell takes a total amount of 200 MCS to divide, provided that it exceeds the threshold size of a solitary reference cell $A_\text{ref}$. Because the first daughter cells may only appear after 200 MCS have passed, we exclude this initial period from the measurements of the maximal front position and roughness, respectively. Additionally, we provide some exemplary time traces of the front evolution (Figure 5-figure supplement B,D,F).

First, we investigated how the monolayer expansion and front roughness depend on cell-substrate dissipation, $\Delta B$ (*Figure 5—figure supplement 1A, B*). Our simulations show that the cell sheet expands slower with increasing cell-cell dissipation $\Delta B$ (*Figure 5—figure supplement 1A, B*), because the dissipation penalizes cells sliding past each other. At the same time, the cell sheet also becomes slightly rougher with increasing cell-cell dissipation $\Delta B$ (inset of *Figure 5—figure supplement 1A*).

We also investigated how the monolayer expansion and front roughness depend on cell-substrate dissipation, $D$ (*Figure 5—figure supplement 1C, D*). Before we turn to the monolayer, let us recall the observed single-cell behavior in the previous section (see section 'Single cell shape and dynamics depend on substrate dissipation'): for high enough cell-substrate dissipation $D$ (typically of the same order of magnitude as the maximum cell polarity

$\Delta\epsilon$) cell migration is switched off (*Figure 2—figure supplement 1*). Extrapolating the single-cell results, we expect that the same holds also for collectives of cells and that cell migration does not play a role for high cell-substrate dissipation. Indeed, with increasing cell-substrate dissipation, the monolayer expands slower, until this effect appears to saturate at a threshold value $D^\star \approx 5$ (*Figure 5—figure supplement 1C, D*). Following this line of argument, monolayer growth is slowed down if we suppress cell migration and thus move the cell monolayer towards a proliferation-dominated mode of expansion.

What about the inverse? Is the monolayer growth increased if we enhance cell migration and thus move the cell monolayer towards a migration-dominated mode of expansion? To test this hypothesis, we have analyzed how the monolayer growth and front roughness depend on the maximum cell polarity $\Delta\epsilon$. As predicted, monolayer growth increases with the maximum cell polarity $\Delta\epsilon$ (*Figure 5—figure supplement 1E, F*), because an increased amount of cells exceed the threshold size to switch to mitosis (cf. the stretching of bulk cells in the monolayer in *Figure 5B*). Additionally, we also find that the front roughness increases with increasing maximum cell polarity $\Delta\epsilon$.

