## [Decision Letter]

[Editors’ note: this article was originally rejected after discussions between the reviewers, but the authors were invited to resubmit after an appeal against the decision.]

Thank you for submitting your work entitled "Bridging the gap between single-cell migration and collective dynamics" for consideration by *eLife*. Your article has been reviewed by three peer reviewers, one of whom is a member of our Board of Reviewing Editors, and the evaluation has been overseen by a Senior Editor.

Our decision has been reached after consultation between the reviewers. Based on these discussions and the individual reviews below, we regret to inform you that your work will not be considered further for publication in *eLife*.

All three reviewers shared concerns that the model was insufficiently connected to real data, and that the novelty of the model with regards to existing models of the same flavour (cellular Potts model) where not sufficiently highlighted.

They all agreed that providing a model that is applicable to both single cell and multi-cellular phenomena is a worthwhile endeavour to "bridge the gap" between single and collective cell migration. However, the single cell part lacks comparison with real data and proposal for experimental check of falsifiable predictions. The multi-cellular aspect was unanimously found more interesting/novel, but much less developed. Explicit suggestions by reviewer #3 on improvements/extensions of the multi-cellular part will likely require quite a bit work and time to be implemented.

Reviewer #1:

This paper describes a Cellular Potts Model for single and collective cell migration.

Single cell motility reproduces the crescent shape of keratocyte-like crawling cells and gives prediction regarding cell size and polarisation.

At the multicellular level, the model reproduces collective rotation of cells confined in circular domains. The authors have discussed this behaviour in a previous paper (Segerer et al., 2015) using a model which I believe is very similar to the present one (or exactly the same?), but which is not detailed in Segerer et al., 2015. Here, there they find an interesting behaviour, with an optimal ratio of polarizability to contractility for persistent collective cell rotation.

Finally, they study the tissue-level dynamics of a typical "wound-healing" assay, and study motility vs. cell proliferation dynamics.

This paper is study situations of direct experimental and biological relevance and reproduces a number of expected results. This is a valuable addition to the already fairly large body of work on the modelling of cell motility. Furthermore, the same model is applicable to single cell motility but also to collective motility in multicellular systems, which is quite interesting.

On the other hand, I found the model quite complicated, with a number of ad-hoc ingredients. This might be reasonable, since the experimental system is complicated as well, and the ingredients are based on known physiological factors, but the extent to which their results are universal is unclear, and little effort is made to address this point.

The comparison with experiments remains qualitative at best, or even anecdotal. For instance, the recent shape of single migrating cell is universally obtained from a number of models. On the other hand, they predict that cell must be large enough to polarise and move persistently, which is highly non-trivial (may be even counter-intuitive?) and could be a crucial test of the model as compared to others but is not discussed further.

I think going beyond these limitations would probably require quite a bit of additional work. The paper as it stands is a very valuable piece of work but might bring more questions than it gives answers. I would recommend enhancing the discussion on the universality of the model and reinforce the discussion of the prediction is relation to experimental finding, and if possible, make that discussion more quantitative.

Reviewer #2:

There is nothing really wrong with this paper and it would make a fine submission to PLoS Computational Biology. Essentially the authors introduce a more complicated version of the cellular Potts model (CPM) to study individual and collective cell motility. The original CPM had several clear deficiencies involving the lack of active cytoskeleton-based driving and the insufficient treatment of adhesion. There have been some remedies for these deficiencies already published but on the whole the model proposed here seems to be a good step forward.

But I do not see sufficient scientific findings that have emerged so far from this new model to warrant acceptance of this paper for *eLife*. The results on single cell motion are not actually compared to any data, for example to keratocyte shapes versus parameters as available from the Theriot lab. Also, although the authors claim that at small R there is a lack of coherence of active protrusion across the cell front (Figure 3D), when this happens in actual cells such as *Dictyostelium* we see much more pronounced stochasticity involving pseudopods of a characteristic scale; as far as I can tell this does not happen in this model. The fact that cells exhibit persistent random walks with correlation times that depend on various parameters is hardly surprising. Other issues regarding single cell motion, such as guidance in the presence of gradients either chemical or mechanical are not presented at all.

Again, I saw nothing strikingly different from previous results for rotating clusters and again no comparison to any experimental data. The model seems to make reasonable predictions but this is not enough for acceptance given the previously published work on this situation.

For tissues the case is somewhat better. Some of the ideas and results here recapitulate what has been seen in other models; this holds for example for the density structure of the population, which in this paper is based on a division rule that directly mimics those used in earlier works. Other aspects appear more novel, such as the reproduction of some the findings regarding wave-like behavior in expanding tissues. But again, this agreement stays at the purely qualitative level, leaving the reader with no specific predictions which could be used to test the validity of the proposed mechanism.

Finally, it was almost impossible to understand the presentation of the model in the main text, with all sorts of symbols and rules appearing out of nowhere. The appendix remedies this problem, but it would almost be better to just describe the pieces qualitatively in the main text instead of undertaking a valiant but in the end failed attempt to include some detailed formulas. In passing, Figure 1 seems to be missing some subfigures.

Reviewer #3:

The manuscript features a coarse-grained description of cell and tissue migration based on the generalization of the cellular Potts model. I think the work contains enough new results to justify publication in *eLife*. However, prior to the publication, the following points need to be addressed. My main concern is that the potential of the computational model is not fully explored.

1) Computational model. The description of the model is rather comprehensive and clear. However, it is necessary to highlight the differences between the current model and earlier CPM approaches to cell motility, e.g. in Kabla, 2012, Doxzen et al., 2013. What is the main difference that enables bridging the single-cell and collective dynamics?

2) Single cell migration. The computational model demonstrates that the cells can polarize and adopt a crescent shape. However, recent experimental studies predicted a phenotype change for the same cell line: a transition from migration to rotation: (i) Lou et al., 2015. (ii) Raynaud F. et al., 2016. This transition was captured in the framework of the phase-field model for slightly higher rates of actin polymerization, Reeves et al., 2018. I am wondering if the CPM model can also capture this transition for stronger driving parameters.

3) The tissue level dynamics. This part of the work is possibly the most innovative and crucial. However, exploration of the tissue dynamics in this section is rather brief. Certainly, the reproduction of the X-pattern is interesting. But many other relevant phenomena can be easily investigated with this method, e.g. instability of the growing front of the cells. Many experiments show that in the course of wound healing, the cell front exhibits a sort of fingering instability. Is this phenomenon captured by the model or the front always remains flat? Another interesting issues are the change of topology and stress distribution at the onset of confluence, and how the number of neighbors changes in the process of cell migration. The CPM model can be contrasted for example to active vertex model where the topological changes are not allowed, DL Barton, S Henkes, CJ Weijer, R Sknepnek, PLoS computational biology 13 (6), e1005569.

[Editors’ note: what now follows is the decision letter after the authors submitted for further consideration.]

Thank you for resubmitting your work entitled "Bridging the gap between single-cell migration and collective dynamics" for further consideration by *eLife*. Your revised article has been evaluated by Naama Barkai (Senior Editor) and two reviewers, one of whom is a member of our Board of Reviewing Editors.

The reviewers found the paper much improved, in particular regarding the way the model is explained and justified. There are some remaining issues that need to be addressed before acceptance, as outlined below:

– The model for cell division (subsection “Tissue growth by cell division”) should be justified better. Division is said to occur stochastically when the cell size reaches a threshold. Here, the size refers to the spread area, not to the cell volume. Why should division be triggered by an increase of spread area. Single cells actually round up and detach from the substrate before dividing. Are there experimental indications that cell division in an epithelium is triggered by an increase of cell area?

– Blebbistatin is claimed not to affect cell shape of speed (subsection “Cell persistence increases with polarizability”), which is explained by it affecting polarizability and contractility to the same extent. This argument is not entirely convincing. First, it is not clear why blebbinstatin should reduce protrusive force due to actin polarisation. One could make the counter argument that myosin contraction reduce protrusion forces by increasing actin retrograde flow, and blebbistatin-treated cells are often seen to spread faster than intreated cells. Second, the effect of blebbistatin on single cell motility is very diverse, and highly depend on the cell type. Both enhancement and decrease of the migration speed are reported in the literature. This paragraph should not give the feeling that the effect of blebbistatin on motility is universal.

– Regarding the correlation between cell shape and speed (subsection “Cell persistence increases with polarizability”), the model seems apt to capture the behaviour of keratocyte (fast and crescent shaped), but many cell types are actually elongated in the direction of their motion. The is presumably due to polarised protrusion forces along the direction of motion with slow cell detachment at the back. While one can hardly expect a single model to capture all phenotypes, could the author speculate and comment why their model is more appropriate for the crescent-shape phenotype.

– Figure 3F suggest two possible correlation between cell shape and direction of motion: the crescent shape, where cell is elongated perpendicular to its direction of motion, and a cell aligned with the direction of motion. This is not discussed in the text, which is unfortunate. What factor determines in which situation the system is, and how does this affect speed and persistence?

– The discussion of tissue level dynamics uncovers several regimes of growth: i.e. driven by the motility of the cells at the tissues edge, or by cell division. It would be interesting to discuss the extent to which these two regimes could be obtained from a continuous model with an effective viscosity (cf. Appendix 1—figure 3) and a driving force due to motility at the edge or a pressure coming from cell division. Such an effective continuous model should be attempted and compared to the numerical results. For instance, judging from Figure 5, the cell front expands almost linearly in time for motility driven, and may be quadratically in division driven. Could such law be discussed and obtained from simple continuous models?

---

## [Author Response]

[Editors’ note: the author responses to the first round of peer review follow.]

[…]Reviewer #1:This paper describes a Cellular Potts Model for single and collective cell migration.Single cell motility reproduces the crescent shape of keratocyte-like crawling cells and gives prediction regarding cell size and polarisation.At the multicellular level, the model reproduces collective rotation of cells confined in circular domains. The authors have discussed this behaviour in a previous paper (Segerer et al., 2015) using a model which I believe is very similar to the present one (or exactly the same ?), but which is not detailed in Segerer et al., 2015. Here, there they find an interesting behaviour, with an optimal ratio of polarizability to contractility for persistent collective cell rotation.Finally, they study the tissue-level dynamics of a typical "wound-healing" assay, and study motility vs. cell proliferation dynamics.This paper is study situations of direct experimental and biological relevance and reproduces a number of expected results. This is a valuable addition to the already fairly large body of work on the modelling of cell motility. Furthermore, the same model is applicable to single cell motility but also to collective motility in multicellular systems, which is quite interesting.

We thank the reviewer for these positive comments. Note that in our earlier work (Segerer et al., 2015) in collaboration with our experimental colleagues (Raedler lab), we specifically refer to the (present) work submitted to “*eLife*” as the primary source of the model and its implementation. In Segerer et al., we only used the numerical code for a very specific application.

On the other hand, I found the model quite complicated, with a number of ad-hoc ingredients. This might be reasonable, since the experimental system is complicated as well, and the ingredients are based on known physiological factors, but the extent to which their results are universal is unclear, and little effort is made to address this point.

We have tried to address and specifically emphasize this point in Box 1 in the revised manuscript, which briefly explains the background to our approach as follows. Cells are complex active objects with interacting mechanical and chemical regulatory components. These regulatory networks, which can be represented by reaction-diffusion systems, are themselves quite complicated. We chose to reduce the complexity by designing a coarse-grained model that captures some (but probably not all the relevant) general features of these reaction-diffusion networks – for instance, positive feedback, pattern size scales, and signaling ranges. The advantage of this modeling approach is that it enables us to identify which of these (Abstract) features are actually crucial for cell migration. Moreover, at present, the detailed knowledge of the molecular processes required for a more refined model seems out of reach.

The comparison with experiments remains qualitative at best, or even anecdotal. For instance, the recent shape of single migrating cell is universally obtained from a number of models.

Please note that our results go far beyond the emergence of crescent-shaped solitary cells (whose universal appearance is interesting in its own right, as discussed below):

1) In the revised section “Cell persistence increases with polarizability”, we now show that there is a correlation between the elongated shape of solitary cells and their persistent migratory behavior. Then, we investigate how these two phenomena depend on parameters like perimeter stiffness, which models actin cortex contractility. We now compare key features of our numerical simulations with experiments on blebbistatin treated keratocytes (Wilson et al., 2010, from the Theriot lab) and keratocyte fragments (Ofer et al., 2011), and recover the following essential trends:

“Then, polarizability and contractility concomitantly depend on the ability of the cell to exert forces, which can be inhibited by blebbistatin treatment. […] Indeed, blebbistatin treatment of keratocytes and keratocyte fragments was reported not to affect cell shape and speed to any significant degree (Wilson et al., 2010; Ofer et al., 2011).”

2) Furthermore, it has recently come to our attention that single keratocytes in an open space can also exhibit circular motion; see e.g. https://doi.org/10.1101/443408. Our model (like some others) also reproduces this migratory mode, as we now show in the revised section entitled “Feedback range determines whether individual cells move persistently or rotate”.

3) It is noteworthy that these various observations are all derived from a coarse-grained model that neither depends on, nor attempts to implement, specifics. Here, cell migration arises from a positive feedback loop that drives both membrane protrusions and retractions. Therefore, we expect that a broad class of models in which a positive feedback drives migration should show qualitatively similar behavior. This might explain why a number of models universally yield similar cell shapes. This is now discussed in the new section “Active self-regulation of the cytoskeleton”.

4) Beyond our single-cell results, our computational model also faithfully reproduces a number of key features of collective cell behavior, including collective rotations of cell clusters in confined geometries (see the section “Cell clusters on circular micropatterns”) and typical stress patterns in expanding tissues (section “Tissue-level dynamics”).

On the other hand, they predict that cell must be large enough to polarise and move persistently, which is highly non-trivial (may be even counter-intuitive?) and could be a crucial test of the model as compared to others, but is not discussed further.

We now discuss this feature in the revised section “Feedback range determines whether individual cells move persistently or rotate”. In our opinion, these results are not counterintuitive for the following reason: One does not expect a reaction-diffusion system in a confined geometry to exhibit pattern formation if the dimensions of the cell are *smaller* than the length scale of those patterns (Hubatsch et al., 2019). To the best of our knowledge, this effect also compares well to experimental results, where cell motility decreases rapidly as the cell density in a tissue rises.

I think going beyond these limitations would probably require quite a bit of additional work. The paper as it stands is a very valuable piece of work but might bring more questions than it gives answers. I would recommend enhancing the discussion on the universality of the model and reinforce the discussion of the prediction is relation to experimental finding, and if possible, make that discussion more quantitative.

We have now substantially extended the discussion on the universal aspects of our computational model in the section “Active self-regulation of the cytoskeleton”, by comparing features of reaction-diffusion-models with our coarse-grained approach.

Reviewer #2:[…]But I do not see sufficient scientific findings that have emerged so far from this new model to warrant acceptance of this paper for eLife.

First of all, we’d like to thank the reviewer for taking the time to prepare such a detailed and comprehensive review. We greatly appreciate the detailed reading of our manuscript. The reviewer’s comments were very insightful, and their questions and suggestions were very helpful for improving the paper.

In the following we will argue that our model does in fact add to the existing literature on computational models for cells in an interesting and highly significant way. We hope that the arguments presented are convincing enough to change the reviewer’s view as to the suitability of our article for *eLife*.

The results on single cell motion are not actually compared to any data, for example to keratocyte shapes versus parameters as available from the Theriot lab.

In the revised section “Cell persistence increases with polarizability” we have investigated the relationships between keratocyte shape, speed and persistence. We find that there is a correlation between elongated cell shape and cell motility (persistence and speed), which we now qualitatively compare to Keren et al., 2008 from the Theriot lab.

Furthermore, in our revised manuscript we have added a comparison of our numerical results to experiments on blebbistatin-treated keratocytes (Wilson et al., 2010, from the Theriot lab) and keratocyte fragments (Ofer et al., 2011). In these experimental studies, the authors argue that blebbistatin treatment of keratocytes and keratocyte fragments has only marginal effects on cell shape and cell speed. In our computational model, blebbistatin treatment would be equivalent to adjusting two model parameters: (1) cell polarizability due to a decrease in the force that a cell can exert on the substrate, and (2) perimeter stiffness due to a decrease of cell contractility. If both cell polarizability and cell contractility have the same functional dependence on myosin activity, then the ratio between polarizability and contractility should remain constant, and our model predicts that neither cell motility nor cell shape will be significantly affected for keratocyte-like cells, which is consistent with the results of both experimental studies.

Also, although the authors claim that at small R there is a lack of coherence of active protrusion across the cell front (Figure 3-D), when this happens in actual cells such as Dictyostelium we see much more pronounced stochasticity involving pseudopods of a characteristic scale; as far as I can tell this does not happen in this model.

Figure 3—video 1 shows that cells with low signaling radius and high polarizability can form pseudopods and move much more randomly (i.e. at high cell speed and with low cell persistence). This is now discussed in the revised section “Feedback range determines whether individual cells move persistently or rotate”.

The fact that cells exhibit persistent random walks with correlation times that depend on various parameters is hardly surprising.

Indeed, this fact is not at all surprising. We hope that it did not come across as a major insight of the analysis. It merely shows that the computational model is able to describe such persistent motion and, more importantly, it reveals which of the model parameters affect the persistence time. The essential insight gained by applying the computational model to single cell motion is the latter.

It has recently come to our attention that single keratocytes in an open space can also exhibit circular trajectories; see e.g. https://doi.org/10.1101/443408. Our model (like some others) also reproduces this migratory mode, as can be seen in the section entitled “Feedback range determines whether individual cells move persistently or rotate”. This feature of single-cell dynamics is not explicitly built into the model and hence constitutes a nontrivial emergent feature. Furthermore, we predict that cells must exceed a certain size in order to migrate at all.

Other issues regarding single cell motion, such as guidance in the presence of gradients either chemical or mechanical are not presented at all.

This is an interesting, but very different topic, which goes far beyond the scope of this paper. In the supplement, we have included a term that represents friction with the substrate, which could be used to model such gradients. Furthermore, note that in a recent manuscript from our group, we consider the substrate as an elastic deformable sheet and discuss how this affects single-cell motion and shape (arXiv:1808.00314 [physics.bio-ph]).

Again, I saw nothing strikingly different from previous results for rotating clusters and again no comparison to any experimental data. The model seems to make reasonable predictions but this is not enough for acceptance given the previously published work on this situation.For tissues the case is somewhat better. Some of the ideas and results here recapitulate what has been seen in other models; this holds for example for the density structure of the population, which in this paper is based on a division rule that directly mimics those used in earlier works.

We thank the Reviewer for pointing out that we may have failed to cite some important previous literature. However, we are not sure which studies the Reviewer is referring to and would appreciate specific citations.

Other aspects appear more novel, such as the reproduction of some the findings regarding wave-like behavior in expanding tissues. But again, this agreement stays at the purely qualitative level, leaving the reader with no specific predictions which could be used to test the validity of the proposed mechanism.

Maybe this point got lost in the previous version of the manuscript. Therefore, we now explicitly state in the revised manuscript:

“Note that the inhomogeneously distributed traction stresses in the monolayer, and its wave-like behavior, ultimately emerge from cell polarization and the ensuing active cell migration. Therefore, these traction patterns would look much less prominent if one were to inhibit cell motility (compare Figure 5C with Figure 5G).”

Finally, it was almost impossible to understand the presentation of the model in the main text, with all sorts of symbols and rules appearing out of nowhere. The appendix remedies this problem, but it would almost be better to just describe the pieces qualitatively in the main text instead of undertaking a valiant but in the end failed attempt to include some detailed formulas. In passing, Figure 1 seems to be missing some subfigures.

We thank the reviewer for this advice. In response we have significantly revised the discussion of the model in the main text to keep it at a more qualitative level. Furthermore, we have added an Abstract graphical representation of the model which tries to illustrate the main features of the model that we have implemented (Figure 1).

Reviewer #3:The manuscript features a coarse-grained description of cell and tissue migration based on the generalization of the cellular Potts model. I think the work contains enough new results to justify publication in eLife. However, prior to the publication, the following points need to be addressed. My main concern is that the potential of the computational model is not fully explored.

First of all, we’d like to thank the reviewer for taking the time to prepare such a detailed and comprehensive review report. We greatly appreciate the detailed reading of our manuscript. The reviewer’s comments were very insightful, and their questions and suggestions were very helpful for improving the paper.

1) Computational model. The description of the model is rather comprehensive and clear. However, it is necessary to highlight the differences between the current model and earlier CPM approaches to cell motility, e.g. in Kabla, 2012; Doxzen et al., 2013. What is the main difference that enables bridging the single-cell and collective dynamics?

We thank the Reviewer for these kind remarks. We have improved Figure 1 to better describe our model. Furthermore, in the revised section on “Active self-regulation of the cytoskeleton”, we now compare our approach to other methods. The basic ingredients of our model (mechanical constraints, together with mechanochemical feedback loops) are similar to those of other models and yield similar phenomenology. What enables us to bridge the scales and investigate large tissues in our model is its simplicity and low computational cost, which scales linearly with cell size and number of cells.

Other CPM approaches to cell motility have implemented a polarization vector that drives the migration of cells. Such approaches typically reproduce cell migration but cannot account for the formation of multiple competing lamellopodia/pseudopods, which would imply a spatially dependent polarization vector field, or, in our case, a scalar field with gradients. We now discuss this point in the revised manuscript:

“Cell migration couples to a single polarity vector (Albert et al., 2016), if propulsive forces are distributed homogeneously throughout the cell. However, this simplification of the former two cases cannot account for the formation of multiple competing lamellopodia/pseudopods.”

2) Single cell migration. The computational model demonstrates that the cells can polarize and adopt a crescent shape. However, recent experimental studies predicted a phenotype change for the same cell line: a transition from migration to rotation: (i) Lou et al., 2015. (ii) Raynaud et al., 2016. This transition was captured in the framework of the phase-field model for slightly higher rates of actin polymerization, Reeves et al., 2018. I am wondering if the CPM model can also capture this transition for stronger driving parameters.

We thank the reviewer for pointing out that keratocytes can also perform rotations. This is indeed very interesting. Our model predicts such a transition from persistent migration to persistent rotations of single cells: see Figure 2—video 1. We observe rotations when the signaling radius is increased. Note that a larger range of mechanochemical feedback is akin to the stronger driving parameters mentioned by the Reviewer. We explain this observation as follows: For high signaling radii (and cells polarizing along their short axis) signaling from the leading and trailing edges results in mutual interference. Therefore, the cell repeatedly adjusts its polarization axis slightly, which leads to rotations. In the revised paper, we have now discussed this feature in section entitled “Feedback range determines whether individual cells move persistently or rotate”.

3) The tissue level dynamics. This part of the work is possibly the most innovative and crucial. However, exploration of the tissue dynamics in this section is rather brief. Certainly, the reproduction of the X-pattern is interesting. But many other relevant phenomena can be easily investigated with this method, e.g. instability of the growing front of the cells. Many experiments show that in the course of wound healing, the cell front exhibits a sort of fingering instability. Is this phenomenon captured by the model or the front always remains flat?

We thank the reviewer for these comments. Indeed, as now discussed in the revised section on “Tissue-level dynamics”, the model can reproduce rougher cell fronts and small cohorts of cells that coherently move into cell-free space, while retaining back-propagating stress waves at the edges of the monolayer and distinct traction force patterns (see Figure 6 and Figure 6—videos 1 and 2). We observe that an increasing mechanical pressure in the monolayer due to the division of cells initiates outward cell migration. Then, cells in the tissue begin to polarize outwards and coordinate their motion with their neighbors, leading to the formation of small coordinated cell cohorts.

To arrive at the corresponding parameter values, we proceeded as follows. We observed that one can slightly increase front roughness if cell polarizability and therefore cell motility is high or if cell-cell dissipation is large (see Figure 5—figure supplement 1). Based on that, we hypothesized that one should observe ‘fingering’ if the parameters are adjusted accordingly, compared to the original parameters used in the wound-healing assay with cell proliferation:

- “Increase of cell motility by decreasing the membrane stiffness and at the same time increasing polarizability and signaling radius of the cells.”

- “Increase of cell-cell dissipation and slight decrease of cell-cell adhesion.”

- “Slower and less homogeneously distributed cell division by increasing the cell threshold size.”

Another interesting issues are the change of topology and stress distribution at the onset of confluence, and how the number of neighbors changes in the process of cell migration. The CPM model can be contrasted for example to active vertex model where the topological changes are not allowed, DL Barton, S Henkes, CJ Weijer, R Sknepnek, PLoS computational biology 13 (6), e1005569.

We thank the reviewer for this suggestion. In fact, our generalized CPM model is able to model that situation as well. However, it seems to us that this goes beyond the scope of the current paper.

[Editors’ note: the author responses to the re-review follow.]

The reviewers found the paper much improved, in particular regarding the way the model is explained and justified. There are some remaining issues that need to be addressed before acceptance, as outlined below:– The model for cell division (subsection “Tissue growth by cell division”) should be justified better. Division is said to occur stochastically when the cell size reaches a threshold. Here, the size refers to the spread area, not to the cell volume. Why should division be triggered by an increase of spread area. Single cells actually round up and detach from the substrate before dividing. Are there experimental indications that cell division in an epithelium is triggered by an increase of cell area?

In response to the reviewers’ comments, we have clarified and improved on the description of the cell division model in the revised manuscript.

From an experimental perspective, it has been observed in multiple studies (Stoker & Rubin, 1967; Pavel et al., 2018, Puliafito, 2012) that cell growth is inhibited by a large cell density, which was coined contact inhibition of proliferation. A large cell density implies that each individual cell has a small spread area. Therefore, we assume that a cell must exceed a certain threshold size to grow. Below this threshold size, cell growth is inhibited (as in our model).

As the reviewers have remarked, in our model we refer to the cell spread area and not to the cell volume. However, in a cell monolayer, an increase in cell volume (mass) due to cell growth should also lead to an increase of cell spread area as the cell not only further expresses additional integrins but also tries to keep the same thickness as the monolayer.

In experiments, cells round up and decrease the substrate contact area before division, while still maintaining some substrate adhesions (Lock et al., 2018). Furthermore, it was shown that adhesion complex area increases during the G1 and S phases of the cell cycle, and then decreases during the G2 phase (Jones et al., 2018). We model cell rounding during cell division (after the initial growth phase) via a depolarization of the cells, assuming that cell division and cell migration are mutually exclusive cell behaviours. Instead of a depolarization, one could also assume a drastic increase of cell contractility (at the cost of an additional parameter), which in addition to cell rounding would also lead to cell detachment from the substrate.

– Blebbistatin is claimed not to affect cell shape of speed (subsection “Cell persistence increases with polarizability”), which is explained by it affecting polarizability and contractility to the same extent. This argument is not entirely convincing. First, it is not clear why blebbinstatin should reduce protrusive force due to actin polarisation. One could make the counter argument that myosin contraction reduce protrusion forces by increasing actin retrograde flow, and blebbistatin-treated cells are often seen to spread faster than intreated cells. Second, the effect of blebbistatin on single cell motility is very diverse, and highly depend on the cell type. Both enhancement and decrease of the migration speed are reported in the literature. This paragraph should not give the feeling that the effect of blebbistatin on motility is universal.

We thank the reviewers for their critique and apologize for having missed some of the literature. Indeed, in (Liu et al., 2010) it was also shown that blebbistatin can increase cell speed. To clarify this point, we have improved our explanation on how blebbistatin might affect cell polarizability:

In our model, the area and perimeter stiffnesses refer to a global and homogeneous contractility of the cell, while the cell polarization field ∈ drives cell migration. Here, the cell polarisation field does not explicitly distinguish between a local extensibility (e.g. due to actin polymerisation), a local contractility (due to myosin contraction) of the cytoskeleton or spatially regulated cell-substrate adhesions. For example, if cell migration is driven by actin polymerization, then blebbistatin treatment will increase protrusion forces by decreasing the retrograde flow. However, if cell migration is driven by myosin contractility, for example by pulling the cell forward or by locally detaching adhesions, then blebbistatin treatment would also decrease cell polarizability.

– Regarding the correlation between cell shape and speed (subsection “Cell persistence increases with polarizability”), the model seems apt to capture the behaviour of keratocyte (fast and crescent shaped), but many cell types are actually elongated in the direction of their motion. The is presumably due to polarised protrusion forces along the direction of motion with slow cell detachment at the back. While one can hardly expect a single model to capture all phenotypes, could the author speculate and comment why their model is more appropriate for the crescent-shape phenotype.

We have now briefly discussed this in footnote 2 in the revised manuscript. In our model, we have implemented a scalar field ∈ that regulates cell migration, and which corresponds to an isotropic stress field. This model shows the tendency to produce keratocyte-shaped migrating cells. This migratory phenomenology corresponds to a persistently migrating force monopole. In (Goychuk et al., 2018), we have shown that a compliant substrate can lead to a transition to an anisotropic force dipole, where the cell is greatly elongated. A similar effect could be achieved by explicitly considering the local orientation of the cytoskeleton. Then, we would have to introduce a nematic field which represents the local orientation of stress fibres (Kassianidou et al., 2019).

– Figure 3F suggest two possible correlation between cell shape and direction of motion: the crescent shape, where cell is elongated perpendicular to its direction of motion, and a cell aligned with the direction of motion. This is not discussed in the text, which is unfortunate. What factor determines in which situation the system is, and how does this affect speed and persistence?

We thank the reviewers for picking up that we have not discussed Figure 3F in sufficient detail. We have amended this issue in the revised manuscript.

– The discussion of tissue level dynamics uncovers several regimes of growth: i.e driven by the motility of the cells at the tissues edge, or by cell division. It would be interesting to discuss the extent to which these two regimes could be obtained from a continuous model with an effective viscosity (cf. Appendix 1—figure 3) and a driving force due to motility at the edge or a pressure coming from cell division. Such an effective continuous model should be attempted, and compared to the numerical results. For instance, judging from Figure 5, the cell front expands almost linearly in time for motility driven, and may be quadratically in division driven. Could such law be discussed and obtained from simple continuous models?

It would indeed be very interesting to reduce our model to an 2D or even 1D model (neglecting shear viscosity) of a compressible active fluid. However, such an endeavour would ultimately require the derivation, implementation and the study of a compressible hydrodynamic model with proliferation. This is significantly beyond the scope of the present manuscript. We speculate that such a model would show the following behaviour. If we suppose that all cells in the monolayer divide, then the total amount of cells should grow exponentially. However, the expansion of the cell front will ultimately be limited by the migration speed of the front cells, which will lead to an increase of cell density over time. Due to contact inhibition of proliferation, such an increase of cell density will lead to an arrest of cell divisions and to a stagnating proliferative ‘pressure’. Therefore, we expect the monolayer expansion velocity to reach a plateau at high cell densities. At low cell densities, the cell speed should increase with time due to feedback mechanisms in the cell polarization.